# N-myc downstream regulated gene 1 (ndrg1) functions as a molecular switch for cellular adaptation to hypoxia

Jong S Park[1], Austin M Gabel[1], Polina Kassir[1], Lois Kang[1], Prableen K Chowdhary[1], Afia Osei-Ntansah[1], Neil D Tran[1], Soujanya Viswanathan[1], Bryanna Canales[1], Pengfei Ding[2], Young-Sam Lee[3], Rachel Brewster[1]*

[1]Department of Biological Sciences, University of Maryland Baltimore County, Baltimore, United States; [2]Department of Chemistry and Biochemistry, University of Maryland Baltimore County, Baltimore, United States; [3]Department of Biology, Johns Hopkins University, Baltimore, United States

**Abstract** Lack of oxygen (hypoxia and anoxia) is detrimental to cell function and survival and underlies many disease conditions. Hence, metazoans have evolved mechanisms to adapt to low oxygen. One such mechanism, metabolic suppression, decreases the cellular demand for oxygen by downregulating ATP-demanding processes. However, the molecular mechanisms underlying this adaptation are poorly understood. Here, we report on the role of *ndrg1a* in hypoxia adaptation of the anoxia-tolerant zebrafish embryo. *ndrg1a* is expressed in the kidney and ionocytes, cell types that use large amounts of ATP to maintain ion homeostasis. *ndrg1a* mutants are viable and develop normally when raised under normal oxygen. However, their survival and kidney function is reduced relative to WT embryos following exposure to prolonged anoxia. We further demonstrate that Ndrg1a binds to the energy-demanding sodium-potassium ATPase (NKA) pump under anoxia and is required for its degradation, which may preserve ATP in the kidney and ionocytes and contribute to energy homeostasis. Lastly, we show that sodium azide treatment, which increases lactate levels under normoxia, is sufficient to trigger NKA degradation in an Ndrg1a-dependent manner. These findings support a model whereby Ndrg1a is essential for hypoxia adaptation and functions downstream of lactate signaling to induce NKA degradation, a process known to conserve cellular energy.

*For correspondence:
brewster@umbc.edu

**Competing interest:** The authors declare that no competing interests exist.

## Editor's evaluation

The manuscript details the function of the N-Myc downstream-regulated gene 1 (NDRG1) during induced hypoxia using the anoxic developing zebrafish as a model system. With some additional support for the central claim of a switch for metabolic suppression, this paper will be of interest to scientists with a focus on kidney development, factors that regulate hypoxic survival, and metabolism in response to stress conditions.

## Introduction

Hypoxia contributes to multiple disease conditions, including acute kidney injury (*Shu et al., 2019*), pulmonary hypertension (*Bosc et al., 2010*), obstructive sleep apnea (*Douglas et al., 2010*), neurodegenerative disease (*Peers et al., 2009*), and ischemia/stroke (*Won et al., 2002*). Despite these hypoxia-related pathological outcomes, low oxygen is encountered in many environments, such as at high altitude, overwintering in frozen ponds, embryonic development *in utero* and in the center of solid tumors. Hence, metazoans have evolved mechanisms to adapt to hypoxia and maintain homeostasis.

These adaptations differ among cell types and species and the degree of protection they confer; however, they generally involve enhanced oxygen delivery, autophagy to recycle cellular constituents, and metabolic reprogramming (switch from oxidative phosphorylation to glycolysis) (*Bellot et al., 2009*; *Guillemin and Krasnow, 1997*; *Semenza, 2001*; *Xie and Simon, 2017*). Important mediators of such responses include hypoxia inducible factor 1α (Hif1α), a transcription factor that is stabilized by low oxygen (*Semenza, 2012*; *Wang and Semenza, 1995*) and AMP-activated kinase (AMPK), whose activation is triggered by an increase in AMP or reactive oxygen species levels (*Dengler, 2020*; *Oakhill et al., 2011*; *Xiao et al., 2011*). In addition, cells adapt to chronic and severe hypoxia via metabolic suppression, a mechanism that preserves energy via reorganization of metabolic priorities for ATP expenditure, suppression of energy-demanding cell functions (e.g. ion pumping, transcription, translation, cell cycle arrest) and changes in gene expression that support hypometabolism (*Storey and Storey, 2007*). In particular, downregulation of ion pumps such as NKA, a major utilizer of cellular ATP, is a key aspect of metabolic suppression (*Bogdanova et al., 2016*; *Buck and Hochachka, 1993*; *Gusarova et al., 2011*). In alveolar cells, this process depends on activation of AMPK (*Gusarova et al., 2011*).

The NDRG family consists of four members (NDRG 1–4) that belong to the α−β hydrolase superfamily, however, they lack the catalytic domain to be enzymatically active (*Shaw et al., 2002*). Several members of this family have altered expression in cancer cells and tumor suppressor or oncogenic functions that impact cell proliferation, differentiation and metastasis (*Melotte et al., 2010*). NDRGs are also implicated in stress response and some members are transcriptionally up-regulated, in a Hif1α-dependent manner, following exposure to hypoxia (*Angst et al., 2006*; *Cangul, 2004*; *Chen et al., 2006*; *Le et al., 2021*; *Melotte et al., 2010*; *Park et al., 2000*). In hypoxic cancer cells, NDRG3 is stabilized by lactate and promotes angiogenesis and cell growth via activation of the Raf-ERK pathway (*Lee et al., 2015*). NDRG1 (formerly known as Drg1, Cap43, Rit42, RTP, and PROXY-1) protects human trophoblasts from hypoxic injury by triggering an anti-apoptotic response (*Chen et al., 2006*; *Choi et al., 2007*; *Roh et al., 2005*). While NDRGs are clearly implicated in the stress response, their physiological roles in hypoxia adaptation of normal cells are mostly unknown.

The zebrafish genome encodes six *ndrg* family members that are highly homologous to mammalian *NDRGs* (*Melotte et al., 2010*). We have previously shown that *ndrg1a* mRNA is up-regulated nine fold in zebrafish embryos exposed to prolonged anoxia (*Le et al., 2021*), despite a general suppression of transcription under severe hypoxia (*Storey and Storey, 2007*). These observations suggest that *ndrg1a* plays an essential role in hypoxia adaptation. *ndrg1a* is primarily expressed in the embryonic kidney (pronephric duct or PD) and ionocytes (*Le et al., 2021*; *Thisse et al., 2001*), cell types that maintain ion homeostasis via the activity of membrane pumps and ion transporters (*Hwang and Chou, 2013*).

We report here on the function of *ndrg1a* in the anoxia-tolerant zebrafish embryo, which can survive up to 50 hr of anoxia in a state of reversible arrest (*Mendelsohn et al., 2008*; *Padilla and Roth, 2001*). While *ndrg1a* is expressed in the PD and ionocytes, it is not required for the differentiation or function of these cells under normal conditions (normoxia). We rather found that *ndrg1a* promotes organismal survival and protects the kidney from hypoxic injury following exposure to prolonged anoxia. Under normoxic conditions, Ndrg1a spatially overlaps with NKA in the PD and ionocytes, but these proteins appear to interact minimally. Following prolonged anoxia, elevated lactate levels (the end product of glycolysis) may be sufficient to enhance Ndrg1a and NKA interaction, which precedes NKA degradation via the lysosomal/proteasomal pathways; a response known to preserve ATP in hypoxia-tolerant organisms. In support of this model, we show that treatment of zebrafish embryos exposed to anoxia in the presence of ouabain, a general NKA inhibitor, rescues ATP levels. We thus propose that Ndrg1a is essential for hypoxia adaptation and functions downstream of lactate signaling to regulate energy-demanding cellular targets, including NKA.

## Results

### *ndrg1a* is not required for kidney and ionocyte differentiation or function

We have previously shown that *ndrg1a* is a hypoxia-responsive gene whose transcript levels increase proportionately with the severity and duration of hypoxia (*Le et al., 2021*), suggesting that *ndrg1a*

is implicated in hypoxia adaptation. However, others have shown that NDRG1 plays a key role in kidney cell differentiation in *Xenopus* (*Kyuno et al., 2003*; *Zhang et al., 2013*), which, if conserved, would preclude directly testing the function of this NDRG family member in cell physiology. We therefore began by investigating whether differentiation of PD cells and ionocytes is impaired in Ndrg1a-deficient zebrafish embryos.

Ndrg1a was targeted using a translation-blocking morpholino (MO) and CRISPR/Cas 9 mutagenesis to produce a null allele, *ndrg1a*mbc1 (*Figure 1—figure supplement 1*). *ndrg1a*mbc1 mutants (referred to henceforth as *ndrg1a*-/-) are homozygous viable and morphologically normal (*Figure 1—figure supplement 1E, F*). Whole-mount *in situ* hybridization (WISH) using several markers (*slc20a1a*: anterior PD, *slc12a3*: posterior PD, *atp1a1a*, the catalytic subunit of NKA: entire length of the PD and ionocytes and *kcnj1a.1*: mid PD and ionocytes), revealed that the distribution of these transcripts is normal in *ndrg1a* loss-of-function (LOF) embryos at 24 hpf (*Figure 1A*). In addition, the expression of *slc20a1a* and *atp1a1a* is properly maintained in 48 hpf *ndrg1a* LOF embryos (*Figure 1—figure supplement 3A*), when the kidney becomes functional (*Drummond et al., 1998*). Furthermore, the lack of signal using sense riboprobes confirmed the specificity of the labeling with the antisense probes (*Figure 1—figure supplement 2*). Immunofluorescence using anti-ATP1A1A and anti-GFP in the transgenic line Tg[*enpep:GFP*] (entire length of the PD) further confirmed proper protein expression in the PD (*Figure 1B*).

Kidney physiology is tightly linked to cell polarity, which enables the directional flow of ions. Analysis of the subcellular distribution of polarity markers F-actin and zonula occludens-1 (ZO-1) showed that *ndrg1a* is not required for apico-basal polarization of PD cells (*Figure 1—figure supplement 3B*).

We next tested the function of the embryonic kidney using a kidney clearance assay (KCA) (*Christou-Savina et al., 2015*). Rhodamine dextran was injected into the pericardial cavity of 48 hpf embryos and the ability of the kidney to clear the rhodamine dextran was tested by measuring fluorescence intensity after 3 and 24 hours post-injection (hpi). This analysis revealed that kidney filtration is normal in *ndrg1a*-/- mutants (*Figure 1—figure supplement 3C, D*).

These findings indicate that *ndrg1a* is neither required for the differentiation of the PD and ionocytes nor for the function of the PD under normoxic conditions, enabling us to directly test the function of *ndrg1a* in physiological adaptation to hypoxia.

## Ndrg1a promotes organismal and cell survival under prolonged hypoxia

To test whether *ndrg1a* promotes hypoxia survival, we exposed 24 hpf embryos to increasing durations of anoxia (6, 12, 18, and 24 hr) and scored viability immediately post-treatment. This analysis showed that WT embryos survived prolonged anoxia exposure somewhat better than *ndrg1a*-/- mutants (18 hr: WT 78.5% vs. *ndrg1a*-/- 52.6%; 24 hr: WT 47.8% vs. *ndrg1a*-/- 23.2%; *Figure 2A*). However, when surviving embryos were returned to normoxia for 2 days, we noticed a dramatic reduction in viability of *ndrg1a*-/- mutants relative to WT embryos (18 hr: WT 79.0% vs. *ndrg1a*-/- 35.5%; 24 hr: WT 58.8% vs. *ndrg1a*-/- 7.1%; *Figure 2B*). Furthermore, a large percentage of the *ndrg1a*-/- mutants that survived anoxia developed mild or severe edema (*Figure 2C*; 6 hr: WT 97.3% no edema, 2.7% mild edema, 0% severe edema vs. *ndrg1a*-/- 69.5% no edema, 30.5% mild edema, 0% severe edema; 12 hr: WT 94.5% no edema, 5.5% mild edema, 0% severe edema vs. *ndrg1a*-/- 65.2% no edema, 28.2% mild edema, 6.6% severe edema; 18 hr: WT 93.1% no edema, 6.9% mild edema, 0% severe edema vs. *ndrg1a*-/- 50% no edema, 41.2% mild edema, 8.8% severe edema). Edema results from water retention and has multiple proximal causes, including malfunction of the PD (*Elmonem et al., 2018*) and ionocytes (*Hwang and Chou, 2013*). We therefore tested whether renal damage occurred post-anoxia using the kidney clearance assay. 24 hpf embryos were subjected to 12 hr of anoxia and 2 days of re-oxygenation, following which rhodamine dextran was injected into their pericardial cavity and fluorescence measurements were made 3 and 24 hpi. This analysis revealed that *ndrg1a*-/- mutants retained more rhodamine dextran than WT embryos post-anoxia (WT: 44.3% fluorescence vs. *ndrg1a*-/- mutants: 60.3% fluorescence; *Figure 2D and E*), indicative of reduced kidney filtration capability.

Together, these data suggest that *ndrg1a* protects the kidney (and possibly ionocytes) from hypoxia-induced cellular damage, which reduces edema formation and enhances survival.

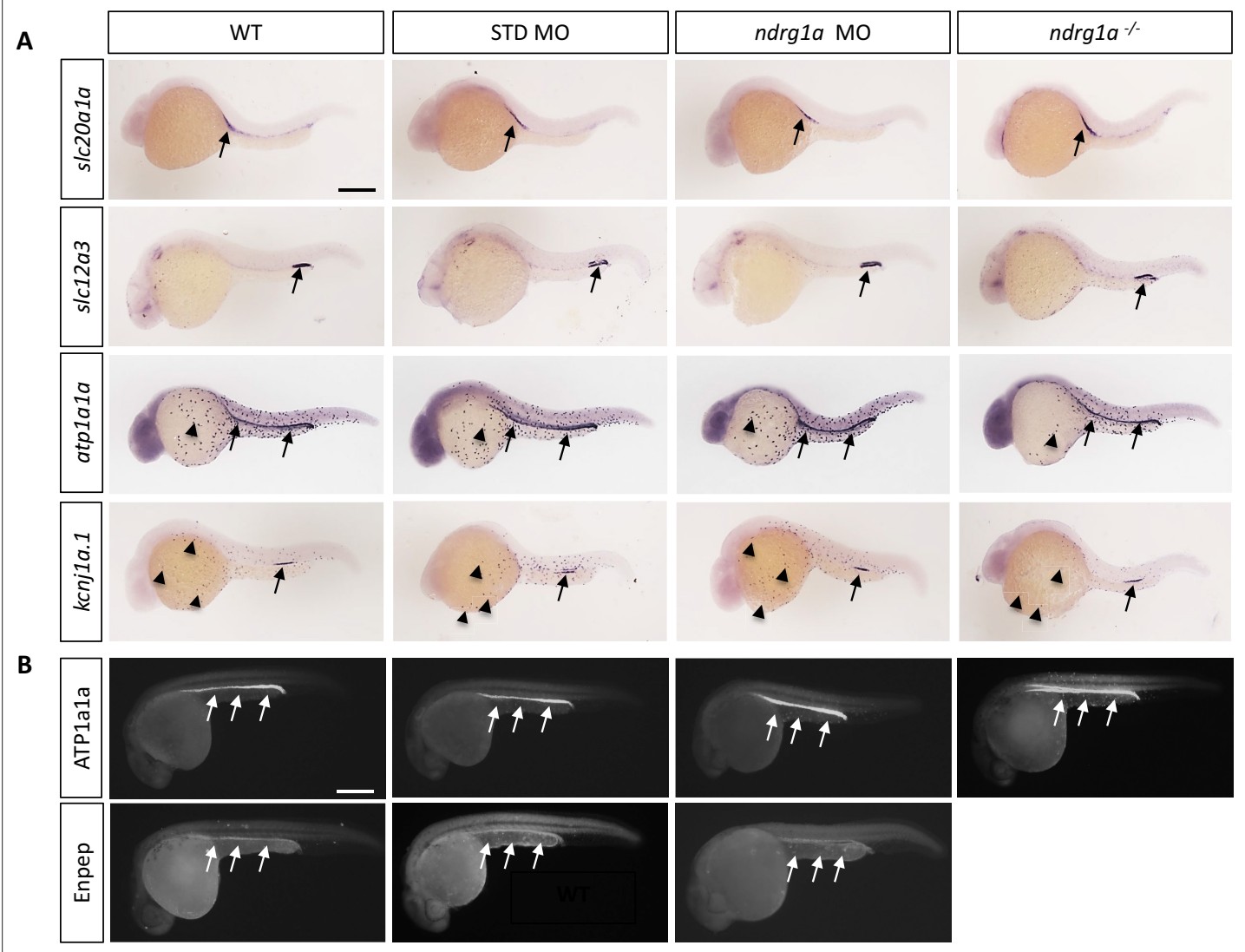

**Figure 1.** Ndrg1a is not required for the differentiation of the pronephric duct and ionocytes. (**A, B**) Lateral views of 24 hpf WT, standard MO-injected, *ndrg1a* MO-injected and *ndrg1a*-/- mutant embryos labeled using (**A**) wholemount *in situ* hybridization to detect the mRNA distribution of *slc20a1a*, *slc12a3*, *atp1a1a, and kcnj1a.1* (n=3–4 experiments with an average of 36 embryos per riboprobe) and (**B**) immunolabeling with anti-ATP1A1A or anti-GFP in Tg[*enpep*:GFP] transgenic line (n=3–4 experiments with an average of 40 embryos per antibody). Annotations: arrows point to signal in the pronephric duct; arrowheads: ionocytes. Scale bars 300 μm.

The online version of this article includes the following source data and figure supplement(s) for figure 1:

**Figure supplement 1.** Generation of *ndrg1a* loss-of-function tools.

**Figure supplement 1—source data 1.** Source data of uncropped, annotated WB figures of Ndrg1a and GAPDH; Raw, unedited WB file for Ndrg1a; Raw, unedited WB file for GAPDH.

**Figure supplement 2.** *In situ* sense probe controls.

**Figure supplement 3.** The morphology, differentiation and function of the pronephric duct are normal in Ndrg1a loss-of-function embryos.

**Figure supplement 3—source data 1.** Source data for quantification of kidney clearance assay under normoxia (48 hpf) WT and *ndrg1a*-/- mutant.

## NKA is downregulated following prolonged anoxia

The expression of *ndrg1a* in the PD and ionocytes and the edema observed in mutants exposed to severe and prolonged hypoxia suggest that Ndrg1a plays a regulatory role, perhaps by controlling ion homeostasis. Of the multiple ion channels and pumps expressed in these cells (*Figure 1*), the catalytic subunit of the NKA pump, ATP1A1A, was selected for further analysis. This enzyme is estimated to consume 40-70% of oxygen-coupled ATP depending on the cell type to maintain Na$^+$ and K$^+$ gradients

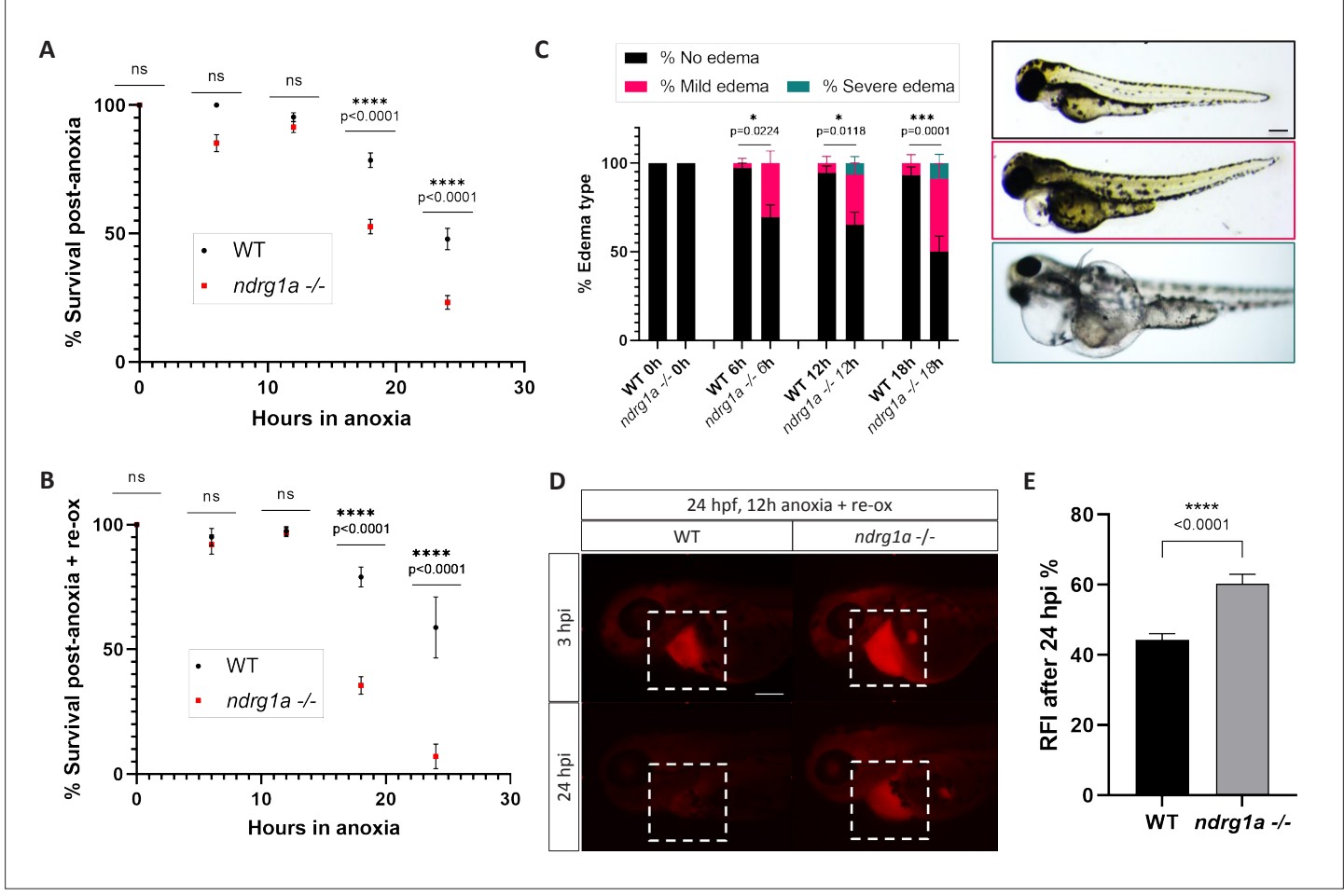

**Figure 2.** *Ndrg1a* confers protection from hypoxic injury. (**A**) % Survival of WT embryos and *ndrg1a⁻/⁻* mutants immediately following 6, 12, 18, and 24 hr of anoxia exposure. Embryos were 24 hpf at the time of exposure. A significant difference between WT and mutants was observed for 12 and 18 hr of anoxia (Two-way ANOVA analysis was performed; 0 hr WT vs. *ndrg1a⁻/⁻*: p-value > 0.9999, ns; 6 hr WT vs. *ndrg1a⁻/⁻*: p-value = 0.0793, ns; 12 hr WT vs. *ndrg1a⁻/⁻*: p-value = 0.9941, ns; 18 hr WT vs. *ndrg1a⁻/⁻*: p-value < 0.0001, ****; 24 hr WT vs. *ndrg1a⁻/⁻*: p-value < 0.0001, ****; n=5–8 experiments with an average of 171 embryos per experimental group; *Figure 2—source data 1*). (**B**) % Survival of WT embryos and *ndrg1a⁻/⁻* mutants following 6, 12, 18, and 24 hr of anoxia exposure and 2 days of re-oxygenation (re-ox). Embryos were 24 hpf at the time of initial anoxia exposure. A significant difference between WT and mutants was observed for 18 and 24 hr of anoxia + 2 re-ox (Two-way ANOVA analysis was performed; 0h+re ox WT vs. *ndrg1a⁻/⁻*: p-value > 0.9999, ns; 6 hr+re ox WT vs. *ndrg1a⁻/⁻*: p-value > 0.9999, ns; 12hr+re ox WT vs. *ndrg1a⁻/⁻*: p-value > 0.9999, ns; 18 hr+re ox WT vs. *ndrg1a⁻/⁻*: p-value < 0.0001, ****; 24 hr+re ox WT vs. *ndrg1a⁻/⁻*: p-value < 0.0001, ****; n=2–5 experiments with an average of 66 embryos per experimental group; *Figure 2—source data 2*). (**C**, left) % Edema observed in WT embryos and *ndrg1a⁻/⁻* mutants following 0, 6, 12, 18, and 24 hr of anoxia exposure and 2 days re-ox. Edema phenotypes were divided into three categories: no edema, mild edema and severe edema. A significant difference between WT and mutants was observed for 6, 12, and 18 hr of anoxia + 2 days re-ox for the presence of either mild or severe edema (Two-way ANOVA analysis was performed; 0 hr+re ox WT vs. *ndrg1a⁻/⁻*: p-value > 0.9999, ns; 6 hr+re ox WT vs. *ndrg1a⁻/⁻*: p-value = 0.0224, *; 12 hr+re ox WT vs. *ndrg1a⁻/⁻*: p-value = 0.0118, *; 18 hr+re ox WT vs. *ndrg1a⁻/⁻*: p-value = 0.0001, ***). (**C**, right) Representative images of embryos in each phenotypic category (n=2–3 experiments with an average of 36 embryos per experimental group; *Figure 2—source data 3*). Scale bar 200 µm. (**D**) Lateral views of injected embryos were taken 3 and 24 hr post-injection (hpi). White boxes indicate where fluorescence measurements were taken. Scale bar 200 µm. (**E**) Quantification of kidney clearance assay using 24 hpf WT embryos and *ndrg1a⁻/⁻* mutants exposed to 12 hr of anoxia, followed by 2 days re-ox and injection with rhodamine dextran. A significant difference between WT and mutants was observed 24 hpi (unpaired t test; p-value < 0.0001, ****). (n=2 experiments with an average of 18 embryos per experiment; *Figure 2—source data 4*).

The online version of this article includes the following source data for figure 2:

**Source data 1.** Source data for % survival of WT embryos and *ndrg1a⁻/⁻* mutants immediately following 6, 12, 18, and 24 hr of anoxia exposure.

**Source data 2.** Source data for % survival of WT embryos and *ndrg1a⁻/⁻* mutants following 6, 12, 18, and 24 hr of anoxia exposure and 2 days of re-oxygenation (re-ox).

**Source data 3.** Source data for % edema observed in WT embryos and *ndrg1a⁻/⁻* mutants following 0, 6, 12, 18, and 24 hr of anoxia exposure and 2 days re-ox.

*Figure 2 continued on next page*

*Figure 2 continued*

**Source data 4.** Source data for kidney clearance assay using 48 hpf WT embryos and *ndrg1a*<sup>-/-</sup> mutants exposed to 12 hr of anoxia, followed by 2 days re-ox and injection with rhodamine dextran.

(*Pirkmajer and Chibalin, 2016*; *Rolfe and Brown, 1997*). Given this high-energy demand, an evolutionarily conserved response of cells exposed to low oxygen is to downregulate NKA activity or to degrade the pump itself (*Bogdanova et al., 2016*; *Buck and Hochachka, 1993*; *Gusarova et al., 2011*).

To address whether Ndrg1a and ATP1A1A interact in the zebrafish embryo, we began by examining their distribution. There are five genes encoding the alpha 1 subunit in the zebrafish genome (*atp1a1a.1-atp1a1a.5*), each with a distinct expression profile (*Blasiole et al., 2002*; *Canfield et al., 2002*). The distribution of *atp1a1a.2* is very similar to that of *ndrg1a*, both of which are expressed in the PD and ionocytes (*Figure 3a1, a2*). Double immunolabeling using anti-NDRG1 and anti-ATP1A1A confirmed that Ndrg1a and ATP1A1A are co-expressed in the kidney and ionocytes (*Figure 3a3*), although a small percentage of ionocytes are positive for only Ndrg1a or ATP1A1A (magnified image *Figure 3a3*, colored boxes in *Figure 4a1*).

We next tested whether NKA is hypoxia-responsive in zebrafish by examining ATP1A1A levels in embryos that were exposed to different durations of anoxia. Under normoxic conditions (0 hr time point), ATP1A1A was distributed along the entire length of the PD but its levels were noticeably higher in the posterior PD (*Figure 3b1*). Following 6 hr of anoxia, the intensity of the signal was reduced throughout the PD, but most noticeably in the anterior region (open arrowheads in *Figure 3b2*; quantified in *Figure 3C*). By 12 hr of anoxia, ATP1A1A levels were further decreased along the full anterior-posterior extent of the PD (open arrowheads in *Figure 3b3*; quantified in *Figure 3C*). Additional timepoints of anoxia (18 and 24 hr) resulted in further NKA downregulation (WT anterior PD: 47.9% reduction of NKA level between 0 hr and 24 hr anoxia; WT posterior PD: 49.1% reduction of NKA level between 0 hr and 24 hr anoxia; *Figure 3C*), but not complete depletion, consistent with prior observations in human and rat lung cells (*Wodopia et al., 2000*). Transverse sections of WT embryos subjected to anoxia and double-labeled with anti-ATP1A1A and the cell surface marker phalloidin (cortical F-actin) revealed that NKA levels were gradually reduced in the baso-lateral plasma membrane compartment of anterior (*Figure 3d1–d3'*, top panels) and posterior (*Figure 3d4–d6'*, bottom panels) PD cells.

ATP1A1A was also downregulated under anoxia in ionocytes that co-expressed Ndrg1a (33.2% reduction of NKA level between 0 hr and 24 hr anoxia), (yellow boxes in *Figure 4*, quantified in *Figure 4B*). In contrast, ionocytes that were Ndrg1a-negative retained high levels of ATP1A1A following 6 and 12 hr of anoxia (green boxes in *Figure 4a1–b3*, quantified in *Figure 4B*). These observations reveal a negative correlation between the presence of Ndrg1a and ATP1A1A levels.

We also observed that the downregulation of ATP1A1A is reversible. Indeed, following 18 hr of anoxia, ATP1A1A levels in both the PD and ionocytes increased steadily when embryos were re-introduced to normoxia (*Figure 3—figure supplement 1A, B*).

Overall, these data indicate that downregulation of NKA is a gradual, long-term response to anoxia that is reversible when oxygen levels return to normal.

## Degradation of NKA is Ndrg1a-dependent

The spatial overlap between Ndrg1a and ATP1A1A and the negative correlation between the levels of these proteins in ionocytes, suggested that Ndrg1a may control NKA abundance under hypoxia. To test this, *ndrg1a*<sup>-/-</sup> mutants were exposed to different durations of anoxia and ATP1A1A levels were quantified. Unexpectedly, analysis of normoxic controls (0 hr anoxia time point) revealed that the PD (*Figure 3b1* vs. b4, quantified in C) and ionocytes (*Figure 4b1* vs. c1, quantified in B) of mutants expressed higher levels of ATP1A1A than cells in WT embryos (58.2% and 20.5% increase in anterior and posterior PD, respectively, of mutant vs. WT), suggesting that Ndrg1a regulates ATP1A1A levels, even under normal oxygen. However, despite these elevated normoxic levels, the relative amount of ATP1A1A in the PD of mutants was only marginally reduced following prolonged anoxia compared to WT embryos in which ATP1A1A levels decreased proportionately to time of anoxia exposure (24 hr anoxia, anterior PD: 8.2% reduction for *ndrg1a*<sup>-/-</sup> vs. 47.9% reduction for WT; 24 hr anoxia, posterior

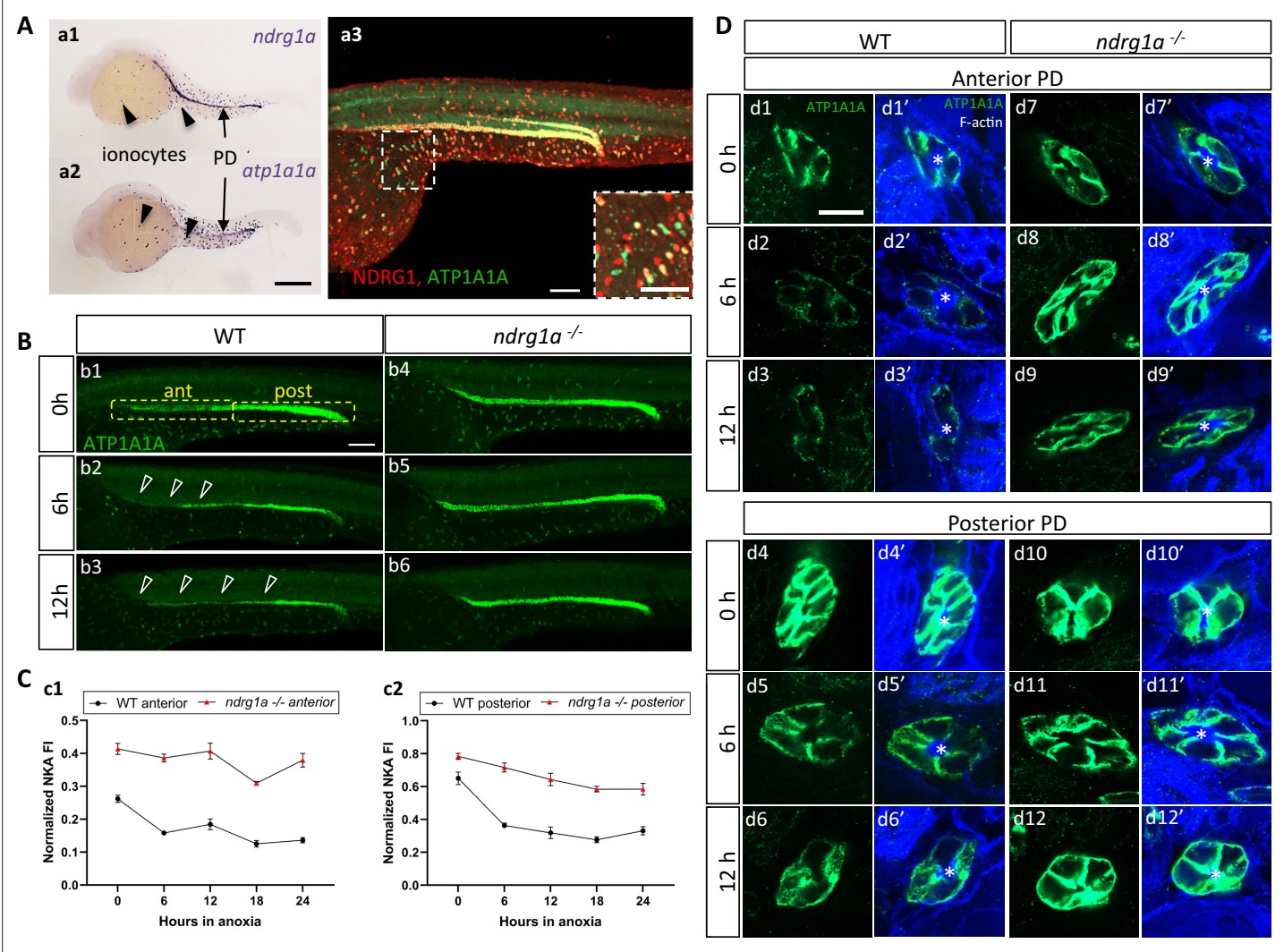

**Figure 3.** NKA downregulatation in the pronephric duct in response to anoxia is *ndrg1a*-dependent. (**A**) Lateral views of 24 hpf WT embryos revealing (**a.1,2**) *ndrg1a* and *atp1a1a* transcript distribution using whole-mount *in situ* hybridization. (**a.3**) Ndrg1a and ATP1A1A protein distribution using double immunolabeling. Scale bars: a1, a2: 300 µm, a3: 100 µm for both wholemount and inset. (**B**) Lateral views of WT embryos and *ndrg1a⁻/⁻* mutants exposed at 24 hpf to increasing duration of anoxia: (**b1, b4**) 0 hr; (**b2, b5**) 6 hr; (**b3, b6**) 12 hr and immunolabeled to reveal ATP1A1A. (n=3 experiments with an average of 29 embryos processed per experiment). Scale bar 100 µm. Abbreviations: ant = anterior, post = posterior. Annotations: yellow boxes show where anterior and posterior pronephric duct measurements were taken. Arrowheads indicate decreased signal. (**C**) Normalized fluorescence intensity in the (**c1**) anterior and (**c2**) posterior pronephric duct of WT embryos and *ndrg1a⁻/⁻* mutants. Embryos were exposed at 24 hpf to increasing duration of anoxia (0, 6, 12, 18, and 24 hr) and immunolabeled to reveal ATP1A1A levels. (n=3 experiments with an average of 21 pronephros segments processed per experimental group; *Figure 3—source data 1*). (**D**) Cross-sections views of the anterior (**d1-d3', d7-d9'**) and posterior (**d4-d6', d10-d12'**) pronephric ducts in WT embryos (**d1-d3', d4-d6'**) and *ndrg1a⁻/⁻* mutants (**d7-d9', d10-d12'**), labeled with anti-ATP1A1A and phalloidin (F-actin). (n=3 experiments with an average of 61 pronephric duct cross-sections processed per experimental group). Scale bar 20 µm. Annotations: asterisks indicate the lumen (apical surface of PD cells).

The online version of this article includes the following source data and figure supplement(s) for figure 3:

**Source data 1.** Source data for normalized fluorescence intensity in the (**c1**) anterior and (**c2**) posterior pronephric duct of WT embryos and *ndrg1a⁻/⁻* mutants.

**Figure supplement 1.** NKA downregulation is reversible upon reoxygenation.

**Figure supplement 1—source data 1.** Source data for quantification of ATP1A1A levels during reoxygenation.

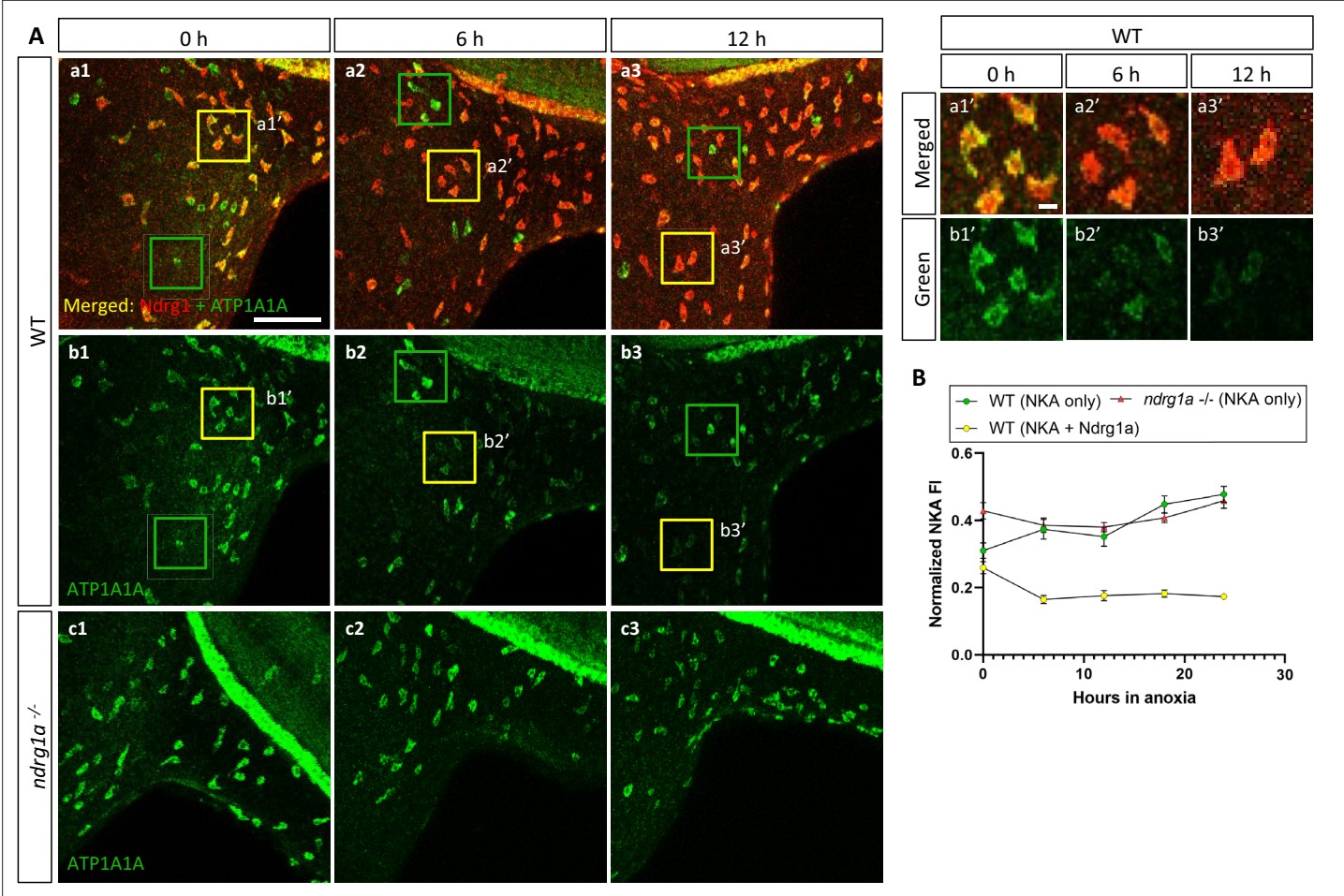

**Figure 4.** NKA is co-expressed with Ndrg1a in a subset of ionocytes and downregulated under anoxia in an *ndrg1a*-dependent manner. (**A, left**) Lateral views of WT embryos (**a1–b3**) and *ndrg1a*[-/-] mutants (**c1–c3**) exposed to anoxia for 0 hr (**a1–c1**), 6 hr (**a2–c2**), 12 hr (**a3–c3**) and immunolabeled with anti-ATP1A1A (green) and anti-Ndrg1a (red); (**a1–a3**) show merged channels, while (**b1–c3**) show the green (ATP1A1A) channel only. Annotations: yellow and green boxes identify ionocytes that are positive for Ndrg1a and ATP1A1A (yellow) or ATP1A1A-only (green). Scale bar: a1: 100 μm; a1′: 2 μm. (**A, right**) Higher magnification images of yellow boxed in areas in (**a1-b3**), showing merged channels (**a1′-a3′**) and green channel only (**b1′-b3′**). (**B**) Normalized fluorescence intensity of ATP1A1A in ionocytes of WT embryos and *ndrg1a*[-/-] mutants exposed to anoxia for 0, 6, 12, 18, and 24 hr. (n=3 experiments with an average of 37 ionocytes processed per experimental group; *Figure 4—source data 1*).

The online version of this article includes the following source data for figure 4:

**Source data 1.** Source data for normalized fluorescence intensity of ATP1A1A in ionocytes of WT embryos and *ndrg1a*[-/-] mutants exposed to anoxia for 0, 6, 12, 18, and 24 hr.

PD: 25.4% reduction for *ndrg1a*[-/-] vs. 49.1% reduction for WT; *Figure 3B*, quantified in C) and even slightly increased in ionocytes of mutants (7% increase; *Figure 4B*).

Furthermore, analysis of cross-sections double-labeled with phalloidin showed that ATP1A1A mostly remained localized in the baso-lateral membrane of PD cells (*Figure 3D*) in *ndrg1a*[-/-] mutants, indicating that the pump is not endocytosed.

These data suggest that Ndrg1a negatively regulates NKA abundance in the plasma membrane under normal oxygen and that this activity is somehow enhanced or accelerated in response to prolonged anoxia.

## Ndrg1a and NKA interact under severe hypoxia

A previous study identified ATP1A1A as a potential binding partner of NDRG1 in human prostate cancer cells (*Tu et al., 2007*), raising the question of whether Ndrg1a physically interacts with ATP1A1A to bring about its downregulation. ATP1A1A is a transmembrane protein, in contrast to

NDRG1 whose peptide sequence does not include an obvious membrane localization signal (*Shi et al., 2013*). Furthermore, NDRG1 intracellular distribution appears to vary between cell types and in response to environmental signals, including hypoxia (*Caruso et al., 2004*; *Kim et al., 2004*; *Kurdistani et al., 1998*; *Lachat et al., 2002*; *Shi et al., 2013*; *Sibold et al., 2007*; *Song et al., 2010*). Given that ATP1A1A downregulation is triggered downstream of hypoxia, in an Ndrg1a-dependent manner, we reasoned that these proteins may co-localize near the cell cortex. This was confirmed in embryos that were double-labeled with anti-NDRG1 and anti-ATP1A1A. Under normoxic conditions, we observed that Ndrg1a was distributed throughout the cytosol and overlapped extensively with ATP1A1A at the cell cortex of ionocytes (*Figure 4a1'*) and PD cells (*Figure 5a1, a4*). The overlap was reduced following 6 and 12 hr of anoxia in both the ionocytes (insets in *Figure 4a2'*, a3') and the anterior PD, coinciding with significant downregulation of ATP1A1A (*Figure 5a2, a3*). In contrast to the anterior PD, the posterior kidney showed extensive overlap of Ndrg1a and ATP1A1A at the cell cortex, even after 12 hr of anoxia (*Figure 5a5, a6*), which is consistent with retention of NKA in the distal kidney tubule (*Figure 3b3, d6*) and indicative of distinct physiological properties of these cells.

To test whether Ndrg1a binds to ATP1A1A, we adapted the PLA for use on whole-mount zebrafish embryos. PLA uses antibodies to two proteins of interest associated with DNA primers; a productive PCR product is produced and detected by fluorescent labeling only when the two antibodies are within less than 40 nm, suggesting that the two target proteins are in close physical proximity and may interact (*Weibrecht et al., 2010*). PLA performed on anoxia-treated embryos using anti-NDRG1 and anti-ATP1A1A antibodies revealed a low signal in the PD for the 0 hr anoxia time point (*Figure 5b1, c1 and c4*), which noticeably increased by 6 and 12 hr of anoxia (*Figure 5b2, b3, c2, c3, c5 and c6*). Cross-sections further showed that the PLA signal in the anterior PD was scattered, localizing to subcortical and possibly intracellular storage compartments (*Figure 5c2 and c3*), whereas it appeared mostly associated with the cell cortex in the posterior PD (*Figure 5c5 and c6*). No PLA signal was detected in *ndrg1a*[-/-] mutants, confirming that the assay is specific for Ndrg1a-ATP1A1A interaction (*Figure 5b4–b6*).

The ability to detect strong PLA signal after 12 hr of anoxia, a time point that correlates with significantly reduced ATP1A1A levels in the PD, is consistent with the high level of sensitivity of this assay and indicates that interaction between Ndrg1a and ATP1A1A is enhanced by hypoxia.

## NKA is downregulated via lysosomal and proteasomal degradation pathways

While ATP1A1A protein levels are not fully depleted following prolonged anoxia, they are greatly reduced. Degradation of transmembrane proteins usually occurs through the lysosome, although these proteins, including potentially NKA, can also be directed to the proteasome (*Helenius et al., 2010*; *Hirsch and Ploegh, 2000*). Hence, blocking lysosomal and/or proteasomal degradation is expected to enhance ATP1A1A levels in embryos exposed to anoxia. To determine the ATP1A1A degradation route, WT embryos were exposed to anoxia (12 hr) and simultaneously incubated in either MG-132 (26 S proteasome inhibitor) or chloroquine (autophagy inhibitor) (*Mathai et al., 2017*) or a combination thereof, and immunolabeled using anti-ATP1A1A. Measurements of fluorescence intensity showed that either drug slightly enhanced ATP1A1A levels in the PD relative to control embryos (12 hr anoxia, no drug) (*Figure 5d2–d4*, quantified in E) (anterior PD: 7.1% increase under 12 hr anoxia + MG-132 vs 12 hr anoxia, no drug; 59.1% increase under 12 hr anoxia + Chloroquine vs. 12 hr anoxia, no drug; posterior PD: 24.4% increase under 12 hr anoxia +MG-132 vs 12 hr anoxia, no drug; 36.9% increase under 12 hr anoxia + Chloroquine vs. 12 hr anoxia, no drug), but the combined use of both drugs had the strongest effect (12 hr anoxia, no drug) (*Figure 5d5*, quantified in E) (anterior PD: 87.4% increase under 12 hr anoxia + MG-132 and Chloroquine vs. 12 hr anoxia, no drug; posterior PD: 45.2% increase under 12 hr anoxia + MG-132 and Chloroquine vs. 12 hr anoxia, no drug). These results suggest that ATP1A1A is partially degraded following prolonged anoxia via both lysosomal and proteasomal pathways. It is possible that Ndrg1a mediates both degradation pathways given that previous studies have identified three subunits of the 26 S proteasome (*Tu et al., 2007*) and lysosomal-associated membrane protein 1 (LAMP1) (*Liu et al., 2018*) as putative NDRG1 binding partners.

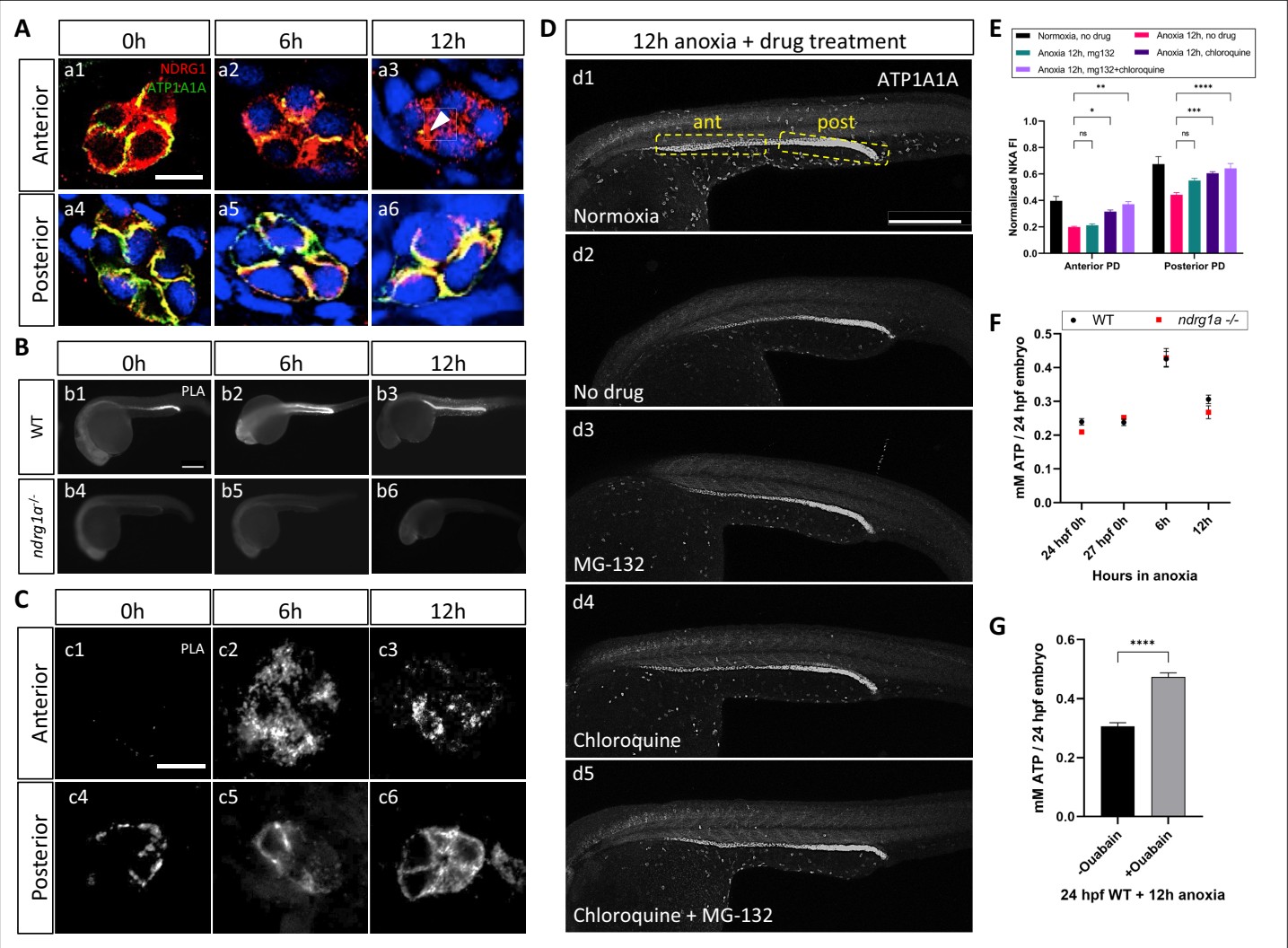

**Figure 5.** NKA interacts with Ndrg1a under anoxia and is degraded via lysosomal and proteasomal pathways to preserve ATP. (**A**) Cross-sections of WT embryos exposed to anoxia for 0 hr (**a1, a4**), 6 hr (**a2, a5**), and 12 hr (**a3, a6**) and immunolabeled with anti-ATP1A1A (green) and anti-Ndrg1a (red). (**a1–a3**) Anterior pronephric duct. (**a4–a6**) Posterior pronephric duct. (n=3 experiments with an average of 33 pronephros sections processed per experimental group). Scale bar = 10 μm. (**B**) Lateral views of WT (**b1–b3**) and *ndrg1a*[-/-] (**b4–b6**) embryos labeled using whole-mount proximity ligation assay (PLA) to reveal Ndrg1a and ATP1A1A interaction. Embryos were exposed to 0 hr (**b1, b4**), 6 hr (**b2, b5**), and 12 hr (**b3, b6**) of anoxia (n=4 experiments with an average of 14 embryos processed per experimental group). Scale bar = 300 μm. (**C**) Cross-sections through the anterior (**c1–c3**) and posterior (**c4–c6**) pronephric duct of WT embryos treated as in (**b1–b3**) (n=2 experiments with an average of 23 pronephros sections processed per experimental group). Scale bar = 10 μm. (**D**) Lateral views of WT embryos exposed to 0 hr (**d1**) or 12 hr of anoxia (**d2–d5**) in presence or absence of MG-132 (100 μM, proteasome inhibitor), or Chloroquine (10 mM, autophagy inhibitor), or both and immunolabeled using anti-ATP1A1A. Annotation: yellow boxes show where anterior and posterior pronephric duct measurements were taken. Scale bar 200 μm. (**E**) Quantification of ATP1A1A fluorescence intensity in the pronephric duct of embryos subjected to the same anoxia and drug treatments as shown in (**D**). (Two-way ANOVA analysis was performed; Anterior PD: 7% increase in Anoxia 12 hr, mg132 vs Anoxia 12 hr, no drug p>0.9999 ns; 59% increase in Anoxia 12 hr, chloroquine vs Anoxia 12 hr, no drug p=0.0298 *; 87% increase in Anoxia 12 hr mg132 + chloroquine vs Anoxia 12 hr, no drug p=0.0013 **. Posterior PD: 24% increase in Anoxia 12 hr, mg132 vs Anoxia 12 hr, no drug p=0.0640 ns, 36% increase in Anoxia 12 hr, chloroquine vs Anoxia 12 hr, no drug p=0.0002 ***, 45% increase in Anoxia 12 hr mg132 + chloroquine vs Anoxia 12 hr, no drug p<0.0001 ****. (n=2 experiments, representative data shown for 1 experiment with an average of 14 pronephros segments processed per experimental group; *Figure 5—source data 1*). (**F**) Quantification of total ATP concentration per WT embryo or *ndrg1a*[-/-] mutant exposed to 0 (24 hpf or 27 hpf developmental stage control), 6, 12 hr of anoxia. No significant difference between WT and mutants were observed. Two-way ANOVA analysis was performed; 0 hr (24 hpf) WT vs. *ndrg1a*[-/-]: p-value = 0.9553, ns; 0 hr (27 hpf developmental stage control) WT vs. *ndrg1a*[-/-]: p value => 0.9999, ns; 6 h WT vs. *ndrg1a*[-/-]: p value => 0.9999, ns; 12 hr WT vs. *ndrg1a*[-/-]: p-value = 0.7652, ns). (n=5 experiments with an average of 15–20 embryos per experimental group; *Figure 5—source data 2*). (**G**) Quantification of total ATP concentration per WT embryo under anoxia either treated with or without 3 mM of ouabain. (Unpaired t test was performed; WT + Ouabain and Anoxia vs. WT + Anoxia only: p value =< 0.0001, ****; *Figure 5—source data 3*).

*Figure 5 continued on next page*

*Figure 5 continued*

The online version of this article includes the following source data and figure supplement(s) for figure 5:

**Source data 1.** Source data for quantification of ATP1A1A fluorescence intensity in the pronephric duct of embryos subjected to 0 hr (**d1**) or 12 hr of anoxia (**d2–d5**) in presence or absence of MG-132, or chloroquine, or both and immunolabeled using anti-ATP1A1A.

**Source data 2.** Source data for quantification of total ATP concentration per WT embryo or *ndrg1a$^{-/-}$* mutant exposed to 0, 6, 12 hr of anoxia.

**Source data 3.** Source data for quantification of total ATP concentration per WT embryo exposed to 12 hr of anoxia with or without ouabain.

**Figure supplement 1.** Isothermal calorimetry (ITC) of WT and *ndrg1a$^{-/-}$* mutant embryos exposed to prolonged anoxia.

**Figure supplement 1—source data 1.** Source data for ITC experiments.

## Inhibition of NKA preserves ATP levels under anoxia

Hypoxia-induced downregulation of NKA is thought to be a key aspect of metabolic suppression, the ultimate goal of which is to preserve cellular energy (*Bogdanova et al., 2016*; *Buck and Hochachka, 1993*; *Gusarova et al., 2011*). To assess whether the activity of Ndrg1a contributes to energy preservation, the total level of ATP (from whole embryo lysates) was quantified using a luminescent ATP assay (*Figure 5F*). We found that ATP levels were similar between WT and *ndrg1a$^{-/-}$* mutants under normoxia (0 hr anoxia time point). Surprisingly, ATP concentration transiently increased in embryos of both genotypes (6 hr anoxia: WT: 79.3% increase; *ndrg1a$^{-/-}$*: 69.8% increase compared to respective 27 hpf stage matched control) before declining after 12 hr of anoxia. Thus, while NKA levels are reduced in the kidney of WT embryos relative to *ndrg1a$^{-/-}$* mutants, this response was not reflected in differential ATP levels. Nevertheless, exposure of embryos to prolonged anoxia (12 hr) in presence of the NKA inhibitor Ouabain, was sufficient to preserve ATP levels at values observed following 6 hr of anoxia (*Figure 5G*).

These results confirm that NKA is a large consumer of ATP and that inhibition of this pump across the entire embryo contributes significantly to energy preservation. We attribute the lack of difference in total ATP levels in WT vs. *ndrg1a$^{-/-}$* mutants to activity of Ndrg1a that is restricted to the kidney and ionocytes. The unexpected transient rise in ATP at 6 hr of anoxia may be due to sustained oxidative phosphorylation, topped by an increase in glycolysis and likely Ndrg1a-independent mechanisms that promote energy conservation.

## Ndrg1a functions downstream of metabolic suppression

Anoxia-induced metabolic suppression in the zebrafish embryo is manifested by developmental arrest, which occurs shortly after anoxia exposure. In contrast, Ndrg1a-mediated NKA degradation is a delayed response that is initiated after several hours of anoxia exposure, suggesting that Ndrg1a signaling may be required to maintain rather than establish hypometabolism. To further test this prediction, we adapted isothermal calorimetry in conjunction with anoxia treatment to measure heat dissipation in whole zebrafish embryos as a readout for metabolic suppression, which is tied to the activity of NKA and other energy-demanding processes. (*Clarke et al., 2013*). Using this approach, we found that heat dissipation is decreased in both WT and *ndrg1a$^{-/-}$* mutant embryos exposed to prolonged anoxia (24 hr), but the percent change was more significant for mutants (69.1%) than for WT (45.4%) embryos (*Figure 5—figure supplement 1C*).

These findings suggest that metabolic suppression is indeed established in both WT and *ndrg1a$^{-/-}$* mutant embryos. However, the lower values observed for mutants are indicative of a disruption in energy homeostasis, possibly linked to deregulation of NKA and other Ndrg1a cellular targets in the kidney and ionocytes.

## Lactate as a candidate regulator of Ndrg1a signaling

Thus far, our data suggest that activation of Ndrg1a accelerates NKA degradation following prolonged anoxia. However, the proximal signal that activates Ndrg1a downstream of anoxia is unknown. We reasoned that this signal could be a small molecule whose levels accumulate following hypoxia exposure. To gain insight into the identity of this putative signal, we carried out metabolite profiling using pre-gastrula zebrafish embryos. Early embryos have a relatively homogenous cell population, facilitating the identification of candidate small molecules. Furthermore, they arrest rapidly following anoxia exposure (*Padilla and Roth, 2001*), which provides a convenient read out for metabolic suppression. We therefore carried out metabolite profiling using 5 hpf WT embryos subjected to

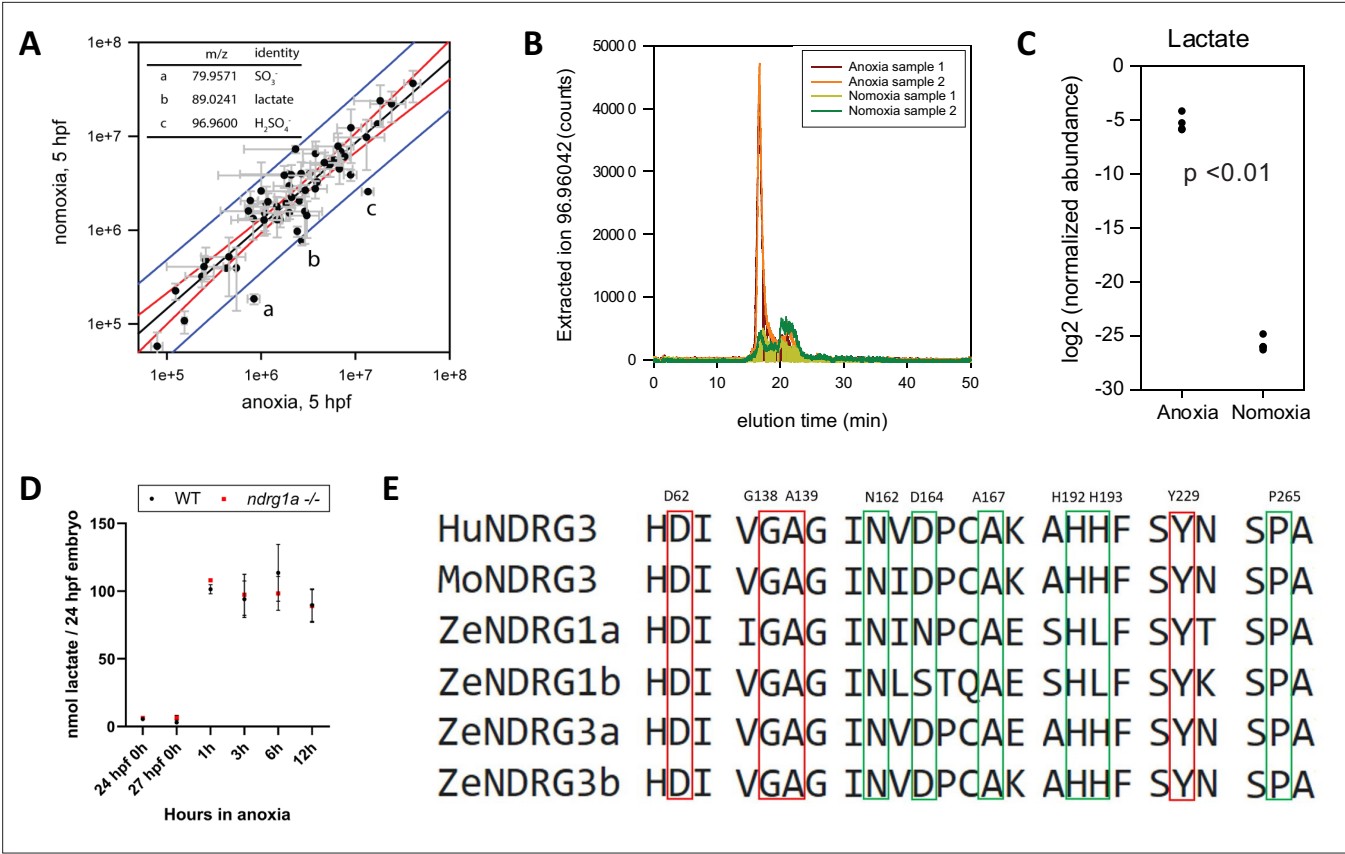

**Figure 6.** Lactate concentration increases under anoxia. (**A graph**) Metabolites enriched in anoxia-treated embryos relative to normoxic controls. X axis: metabolite levels (integrated ion counts) in extracts from 5 hpf embryos exposed to 1 hr of anoxia; Y axis: metabolite levels (integrated ion counts) in extracts from age-matched 5 hpf normoxic controls. The red lines show the predicted correlation line (99% confidence). Blue lines demarcate outliers (99% confidence). Three potential outliers (**a, b and c**) were identified (n=10 embryos per sample and 2 technical repeats). (**A, insert, top left**) Identity of the three outliers (**a–c**) that are significantly enriched in the anoxia-treated sample. (**B**) Extracted ion counts for different elution timepoints are shown for two extracts from 5 hpf embryos exposed to 1 hr of anoxia (orange and red lines) and two control samples from 5 hpf normoxic embryos (yellow and green lines). The peak corresponds to lactate. (**C**) Comparison of lactate levels (highest peak intensity) in extracts from 5 hpf embryo exposed to 1 hr of anoxia and 5 hpf normoxic controls reveals an 18 fold increase (unpaired t test; p<0.01);Figure 6—source data 1A-C . (**D**) Fluorometric lactate assay quantification of total lactate concentration (in nmol lactate) per WT embryo or *ndrg1a*-/- mutant exposed to 0 (24 hpf or 27 hpf developmental stage control), 1, 3, 6, 12 hr of anoxia. No significant difference between WT and mutants were observed. Two-way ANOVA analysis was performed; 0 hr (24 hpf) WT vs. *ndrg1a*-/-: p value => 0.9999, ns; 0 hr (27 hpf developmental stage control) WT vs. *ndrg1a*-/-: p value => 0.9999, ns; 1 hr WT vs. *ndrg1a*-/-: p value => 0.9999, ns; 3 hr WT vs. *ndrg1a*-/-: p value => 0.9999, ns; 6 hr WT vs. *ndrg1a*-/-: p value = 0.9870, ns; 12 hr WT vs. *ndrg1a*-/-: p value => 0.9999, ns. (n=4 experiments with an average of 15 embryos per experimental group; *Figure 6—source data 2*). (**E**) Alignment of members of the NDRG family revealing lactate-docking residues, identified in NDRG3 by *Lee et al., 2015*. Green and red residues represent amino acids in NDRG3 that are 5 ångström from lactate. Red residues represent amino acids that may be critical for the interaction between NDRG3 and lactate.

The online version of this article includes the following source data for figure 6:

**Source data 1.** Source data for *Figure 6*.

**Source data 2.** Source data for fluorometric lactate assay showing lactate concentration (nmol lactate/embryo) in whole embryo extracts from 24 hpf WT or *ndrg1a*-/- mutant exposed to 0, 1, 3, 6, 12 hr of anoxia.

1 hr of anoxia along with age and developmental stage-matched controls raised under normoxic conditions. Following extraction of metabolites, mass spectrometry was performed to identify small molecules whose levels are either up- or downregulated (*Figure 6A–C*). This analysis revealed three candidate ions: sulfite, sulfate, and lactate, which lies outside of the 99th percentile confidence interval for outliers (a-c in *Figure 6A*). We chose to further pursue lactate, given that this metabolite, the end product of glycolysis, was previously shown to bind to NDRG3 in hypoxic cancer cells, resulting in NDRG3 stabilization and activation of Ras-Erk signaling (*Lee et al., 2015*).

Lactate accumulation in anoxic samples was verified by measuring HILIC LC-MS extracted ion counts (*Figure 6B*) and quantified using integrated ion count areas (*Figure 6C*). To test whether an increase in L-lactate also occurs in older 24 hpf embryos and determine the temporal profile of lactate accumulation, we performed a lactate fluorometric assay (*Figure 6D*). This analysis revealed that lactate concentration increased following anoxia, consistent with previous observations (*Podrabsky et al., 2007*; *Vianen et al., 2001*; *Virani and Rees, 2000*). This conserved response reflects a shift from oxidative phosphorylation to anaerobic respiration.

## Inhibition of oxidative-phosphorylation increases lactate and induces NKA degradation

A protein alignment of members of the human, mouse and zebrafish NDRG family revealed that the NDRG3 amino acid residues previously implicated in lactate binding (*Lee et al., 2015*) are conserved across members of this family (*Figure 6E*), suggesting that lactate may also bind to and possibly activate NDRG1.

We next asked whether lactate is sufficient to promote NKA degradation under normoxic conditions. Sodium azide, an electron transport chain inhibitor, was used to increase endogenous lactate levels as previously shown (*Oginuma et al., 2017*) and was confirmed to be effective (*Figure 7A*). We then tested whether exposure of WT embryos to sodium azide (6, 12, 18, 24 hr duration) under normal oxygen caused a decrease in NKA levels (*Figure 7B*). This analysis revealed that NKA levels were indeed reduced in response to sodium azide treatment (WT anterior PD: 58.9% reduction in ATP1A1A between 0 hr and 24 hr anoxia; WT posterior PD: 56.9% reduction in ATP1A1A between 0 hr and 24 hr anoxia; *Figure 7C*). If Ndrg1a is activated downstream of sodium azide, NKA levels should not decrease in *ndrg1a*[-/-] mutants exposed to this drug. Analysis of sodium azide-treated *ndrg1a*[-/-] mutants immunolabeled with anti-ATP1A1A (NKA) revealed that NKA levels were comparably higher in mutants than in WT embryos for the same treatment groups (*ndrg1a*[-/-] anterior PD: 11.7% reduction in ATP1A1A between 0 hr and 24 hr anoxia; *ndrg1a*[-/-] posterior PD: 4.7% reduction in ATP1A1A between 0 hr and 24 hr anoxia; *Figure 7C*), corroborating with this prediction.

Consistent with the ability of sodium azide to downregulate NKA under normoxic conditions, we also observed that azide-treated embryos exhibited enhanced Ndrg1a-NKA interaction, as PLA labeling increased proportionately with the duration of the treatment (*Figure 7D, d1-4*; quantified in E). In contrast, *ndrg1a*[-/-] mutants subjected to the same treatment had no PLA signal, confirming the specificity of this assay (*Figure 7d5*). These data are consistent with lactate being a proximal signal that contributes to NKA downregulation by promoting Ndrg1a-NKA interaction.

## Discussion

Increasing evidence indicates that members of the NDRG family of α−β hydrolases are implicated in the hypoxia response (*Melotte et al., 2010*). However, the mechanisms by which they function in the physiological adaptation of normal cells to low oxygen are, for the most part, unknown. We uncover here a novel and essential role for Ndrg1a in promoting long-term adaptation to hypoxia possibly via metabolic regulation. Overall, our findings support a model whereby prolonged and severe hypoxia activates or enhances Ndrg1a signaling in the zebrafish embryo downstream of lactate. Activated Ndrg1a accelerates the rate of endocytosis and degradation of the energy-demanding NKA pump in the PD and ionocytes, which may preserve ATP. The high level of homology between NDRG1 in vertebrates along with evidence pointing towards conserved molecular interactions, suggest that the proposed model for Ndrg1a signaling may be broadly applicable.

### Ndrg1a as a molecular switch for long-term adaptation to hypoxia

Given that the molecular cascade leading to ATP1A1A internalization and degradation is triggered by prolonged anoxia and dependent on *ndrg1a*, we propose that Ndrg1a functions as a molecular switch for long-term adaptation to hypoxia. The mechanisms underlying Ndrg1a activation have not been fully elucidated, however two non-mutually exclusive models can be envisaged, which we have coined quantitative vs. qualitative. In the quantitative model, Ndrg1a normoxic activity is enhanced in response to hypoxia, possibly via post-translational modifications (PTM) and/or transcriptional up-regulation of *ndrg1a* (*Le et al., 2021*). Consistent with this model, we have shown that *ndrg1a* is

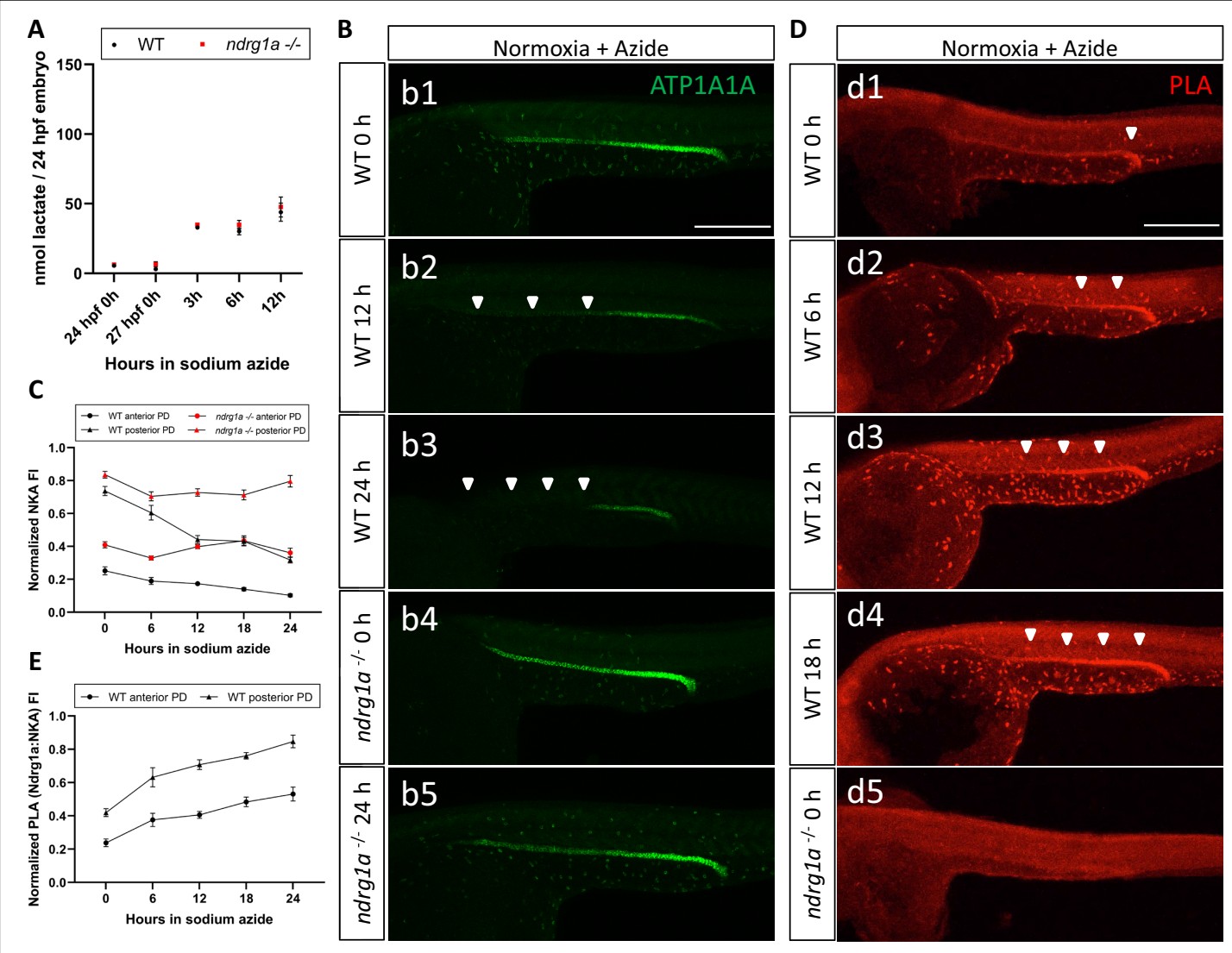

**Figure 7.** Lactate concentration increases following sodium azide treatment under normoxia and correlates with ATP1A1A downregulation. (**A**) Fluorometric lactate assay quantification of total lactate concentration (in nmol lactate) per WT embryo or *ndrg1a*−/− mutant exposed to 0 (24 hpf or 27 hpf developmental stage control), 3, 6, 12 hr of 300 uM of azide. No significant difference between WT and mutants were observed. Two-way ANOVA analysis was performed; 0 hr (24 hpf) WT vs. *ndrg1a*−/−: p value => 0.9999, ns; 0 hr (27 hpf developmental stage control) WT vs. *ndrg1a*−/−: p-value = 0.9994, ns; 3 hr WT vs. *ndrg1a*−/−: p value => 0.9999, ns; 6 hr WT vs. *ndrg1a*−/−: p-value = 0.9919, ns; 12 hr WT vs. *ndrg1a*−/−: p-value = 0.9986, ns. (n=4 experiments with an average of 15 embryos per experimental group; *Figure 7—source data 1*). (**B**) Lateral views of WT embryos and *ndrg1a*−/− mutants exposed at 24 hpf to increasing duration of azide under normoxia: (**b1-5**); 0 hr (**b1, b4**), 12hr (**b2**), 24 hr (**b3, b5**) and immunolabeled to reveal ATP1A1A. Arrowheads indicate decreased signal. (n=3 experiments with an average of 10 embryos processed per experimental group). Scale bar 300 µm. (**C**) Normalized NKA fluorescence intensity in the anterior (circle) and posterior (triangle) pronephric duct of WT (black) embryos and *ndrg1a*−/− (red) mutants. Embryos were exposed at 24 hpf to increasing duration of azide under normoxia (0, 6, 12, 18, and 24 hr) and immunolabeled to reveal ATP1A1A levels (n=2 experiments with an average of 19 pronephros segments processed per experimental group; *Figure 7—source data 2*). (**D**) Lateral views of WT embryos (**d1-4**) and an *ndrg1a*−/− mutant (**d5**) labeled using whole-mount proximity ligation assay (PLA) to reveal Ndrg1a and ATP1A1A interaction. Embryos were raised under normoxic conditions in the presence (6, 12, 18, 24 hr exposure) or absence of sodium azide. Annotations: arrowheads point to PLA signal (n=4 experiments with an average of 10 embryos processed per experimental group). Scale bar 300 µm. (**E**) Normalized PLA intensity in the anterior (circle) and posterior (triangle) pronephric duct of WT embryo. (n=2 experiments with an average of 16 pronephros segments processed per experimental group; *Figure 7—source data 3*).

The online version of this article includes the following source data for figure 7:

**Source data 1.** Source data for fluorometric lactate assay showing lactate concentration (nmol lactate/embryo) in whole embryo extracts from 24 hpf WT or *ndrg1a*−/− mutant exposed to sodium azide for 0, 3, 6, 12 hr under normoxia.

**Source data 2.** Source data for normalized NKA fluorescence intensity in the anterior (circle) and posterior (triangle) pronephric duct of WT (black)

*Figure 7 continued on next page*

*Figure 7 continued*

embryos and *ndrg1a⁻/⁻* (red) mutants.

**Source data 3.** Source data for normalized PLA intensity in the anterior (circle) and posterior (triangle) pronephric duct of WT embryo.

required to maintain normoxic levels of NKA in the plasma membrane and to accelerate NKA degradation under hypoxia. Others have reported on similar NDRG1-dependent regulation of LDLR and E-Cadherin levels under normal oxygen (*Kachhap et al., 2007*; *Pietiäinen et al., 2013*), however it is unknown whether such activities are enhanced under hypoxia.

In the qualitative model, Ndrg1a acquires new properties under hypoxia due to a PTM or binding to a small molecule, which brings about a change in protein conformation and/or binding partner affinity. In support of this model, a previous study showed that binding of lactate to NDRG3 in hypoxic cancer cells is sufficient to stabilize NDRG3 and activate Raf-ERK signaling (*Lee et al., 2015*). We have shown that sodium azide treatment, which blocks oxidative-phosphorylation and increases the intracellular concentration of lactate, is sufficient to induce ATP1A1A degradation under normoxic conditions in an Ndrg1a-dependent manner. While several metabolites could function downstream of sodium azide as effectors of this response, our data point to lactate as a prime candidate.

There is currently no consensus on the molecular role of activated NDRG1 (*Melotte et al., 2010*), however we favor the idea that Ndrg1a functions as an adapter protein to link protein-binding partners together. Indeed, there is a growing list of NDRG1 binding partners that localize to several cellular compartments (*Tu et al., 2007*). Our findings suggest that one of the protein complexes that NDRG1 could bring together under hypoxic conditions includes ATP1A1A, endosomal components and/or protein degradation machinery.

## Hypoxia-induced NKA downregulation as a mechanism to preserve ATP

The NKA isoforms that are expressed in the kidney are essential for the active translocation of $Na^+$ and $K^+$ ions across the membrane and in the secondary active transport of other solutes, including glucose (*Katz, 1982*). The cost of operating NKA is considerable, and hence an evolutionarily conserved response to hypoxia is to downregulate this pump to conserve ATP. Mechanisms of downregulation include reducing pump activity (inactivation via several redox-sensitive modifications) and/or triggering its endocytosis and degradation (*Bogdanova et al., 2016*; *Buck and Hochachka, 1993*; *Gusarova et al., 2011*). We confirm that NKA inhibition, via Ouabain treatment, preserves ATP in zebrafish embryos exposed to prolonged anoxia. Whether Ndrg1a-mediated NKA degradation conserves energy in kidney and ionocytes is less clear, as tools to measure ATP levels in specific tissues and cells are currently lacking.

To what extent does Ndrg1a-mediated degradation of NKA contribute to hypometabolism? Energy preservation via metabolic suppression is a relatively rapid and organism-wide response, resulting in developmental arrest in early embryos. In contrast, Ndrg1a activation is slow (taking place over several hours) and spatially restricted, suggesting that it is a tissue-specific, last resort response to preserve the kidney and ionocytes. Consistent with the delayed and localized activity of Ndrg1a, we observed that *ndrg1a⁻/⁻* mutants are largely able to establish metabolic suppression. However, energy homeostasis appears to be disrupted in the mutants, aligning with the low survival data following prolonged anoxia (24 hr). We thus propose that Ndrg1a maintains energy homeostasis by reducing energy usage in the kidney and ionocytes.

In order for NKA downregulation to be considered a hypoxia adaptation, it would have to be reversible upon return to normoxia, which we have confirmed. The cellular source of the post-anoxia protein is unknown, but may involve intracellular stores of endocytosed NKA and/or protein synthesized *de novo*. The former could provide an immediate source of NKA for rapid restoration of pump activity, while the latter, a more delayed response, could fully replenish the cell's NKA supply.

## Context-specific function of NDRG1

Our data support a physiological role for *ndrg1a* in hypoxia-induced NKA degradation. The NKA pump plays an important role in freshwater teleosts such as zebrafish. These fish are faced with the challenge of potential over-hydration and salt depletion, since their body fluids are hyperosmotic with respect to the environment. To compensate for the tendency towards water influx, zebrafish

excrete large volumes of water and limit NaCl loss via reabsorption of $Na^+$ and $Cl^-$ in the kidney distal (collecting) duct and bladder, and also through ionocyte-dependent absorption of these ions from the environment (*Takvam et al., 2021*). Consistent with these findings, zebrafish NKA is expressed more highly in the posterior PD, where its activity may power the bulk of $Na^+$ reabsorption. PLA further revealed regional differences in the manner in which NKA is regulated in the anterior and posterior PD, as ATP1A1A and Ndrg1a appear to interact in posterior cells even under normoxic conditions. It is possible that, owing to the high level of activity of NKA in the posterior tubules, there is a build up of lactate in these cells. If correct, constitutive Ndrg1a-ATP1A1A interaction in posterior cells may prime them for a more effective response to hypoxia.

Studies in amphibians show that NDRG1 controls cell differentiation of endodermal organs (*Kyuno et al., 2003*; *Zhang et al., 2013*), with no indication of a physiological role for this protein. It is unclear what accounts for the difference in NDRG1 function between teleosts and amphibians, but one possibility is the increased number of *Ndrg* paralogs in zebrafish (6 NDRGs in zebrafish versus 4 in *Xenopus laevis*). This increase might have enabled functional diversification of some members of the NDRG family and for *ndrg1a* in particular to evolve a role in hypoxia adaptation. However, there is ample evidence in mouse and human cells that NDRG1 is hypoxia-responsive (*Melotte et al., 2010*), making this an unlikely explanation. Alternatively, *ndrg1a* may be required for PD and iono-cyte differentiation but its function is masked by other members of the NDRG family that compensate for *ndrg1a* LOF.

## NDRG1 and ion homeostasis

While NKA degradation preserves ATP, the loss of this pump can cause a rise in intracellular $Na^+$ concentration and subsequent water influx, culminating in cellular edema (*Bogdanova et al., 2005*; *Leaf, 1956*; *Mudge, 1951*). However, hypoxia-tolerant organisms are thought to counter NKA depletion with a general reduction in membrane permeability via blockage of leak and water channels (*Boutilier, 2001*; *Doll et al., 1991*; *LaMacchia and Roth, 2015*; *Lutz and Nilsson, 1997*). The underlying mechanisms for this phenomenon, known as channel arrest, have remained controversial, especially with regards to the identity of the proximal signal that triggers channel arrest (*Buck and Hochachka, 1993*; *Hochachka and Lutz, 2001*).

PD cells and ionocytes in zebrafish embryos exposed to prolonged anoxia do not show overt signs of cellular edema (*Figures 3D and 4A*), suggesting that channel arrest may apply to this organism. We further speculate that NDRG1 may negatively regulate ion transporters in zebrafish embryos, in addition to mediating NKA degradation. Indeed, *ndrg1a$^{-/-}$* mutants exposed to prolonged anoxia develop severe edema. Moreover, NDRG1 is known to interact with ion transport regulator FXYD (*Arystark-hova et al., 2017*; *Huttlin et al., 2021*; *Yang et al., 2016*) and the level of several ion transporters in cichlids exposed to high salinity was shown to be inversely proportional to that of NDRG1 (*Kültz et al., 2013*). If correct, the implication of this prediction is that NDRG1 could function as a master switch that coordinates both metabolic suppression/energy preservation (NKA downregulation) and ion homeostasis (reduction in membrane permeability).

Given that human NDRG1 is expressed in the kidney as well as other metabolically-demanding tissues (*Melotte et al., 2010*) and interacts with NKA and ion transporter regulator, it is intriguing to consider whether NDRG1-lactate signaling could be modulated to mitigate the cellular damage caused by hypoxic injury.

## Methods
### Zebrafish husbandry and use

Wild type (WT) zebrafish (*Danio rerio*) of the AB strain were reared and manipulated using protocols approved by the Institutional Animal Care and Use Committee at the University of Maryland Baltimore County. Fish were maintained in UV-irradiated, filtered water and exposed to a 12:12 light/dark cycle. Male and female fish were separated by a partition that was removed after the first light to initiate spawning. Embryos were collected and staged according to previously described methods (*Kimmel et al., 1995*). Both age and stage-matched controls were used for hypoxia/anoxia treated embryos. The sex of the embryos used is unknown.

## CRISPR/Cas9 mutagenesis to generate *ndrg1a* mutants

### Target Selection

The online bioinformatics tool CHOPCHOP (https://chopchop.cbu.uib.no/) was used to identify three unique CRISPR targets for *ndrg1a* (NM_001128353, isoform 1; GRCz10). Each guide RNA (gRNA) target sequence was chosen based on homology to other sequences in the published zebrafish genome, predicted off-target effects, and self-complementation. Two gRNAs were used to target the intron-exon splice junctions of *ndrg1a* exons 4 (gRNA1) and 5 (gRNA2), and a third gRNA was complementary to coding sequence in exon 7 (gRNA3) (*Figure 1—figure supplement 1A*).

The CRISPR target sequences are as follows: gRNA1: *5'-CCACGTCATGTTCCTACACGGGG-3'* gRNA2: *5'-AGGCAATGTAACTCACCCAGTGG-3'* gRNA3: *5'-CTTAGAAAGGATGTAAGCACCGG-3'*.

### Generation of insertion-deletions (indels)

Embryos were mutagenized using active ribonucleoproteins (RNPs) consisting of the Alt-R CRISPR/Cas9 high fidelity nuclear-targeted Cas9 protein (HF-nCas9) and synthetic gRNAs according to the manufacturer's recommendations (Integrated DNA Technologies; IDT). RNPs were co-injected in one-cell stage embryos in the animal pole for multiplexed editing of *ndrg1a*.

### CRISPR mutant identification and characterization of the ndrg1a$^{mbc1}$ allele

Genomic DNA sequencing of one of the F2 females identified 1–9 bp deletions in the vicinity of each of the gRNAs (*Figure 1—figure supplement 1B*). Sequencing of the corresponding transcript revealed a large deletion spanning exons 4–15 (likely caused by gRNA1) (*Figure 1—figure supplement 1C*), which resulted in a truncated protein product that lacks the α–β hydrolase fold in addition to the C-terminal three tandem repeats and the phosphopantetheine sequence which are characteristic of NDRG1 (*Melotte et al., 2010*; *Figure 1—figure supplement 1D*). A homozygous mutant line was established for this allele, *ndrg1a$^{mbc1}$* (designated as *ndrg1a$^{-/-}$*). Immunolabeling and western blotting using anti-NDRG1 (directed against AA 82–203 of human NDRG1; UniProt accession no. Q92597) did not detect any signal in *ndrg1a$^{-/-}$* mutants (*Figure 1—figure supplement 1E, G*). In contrast, wholemount *in situ* hybridization (riboprobe targeting the 3' end of the gene and 3'UTR) revealed that the transcript is expressed at normal levels (*Figure 1—figure supplement 1F*). The failure to detect protein is therefore likely to be a consequence of protein truncation.

## Morpholino-mediated Ndrg1a knockdown

Morpholinos (MOs) targeting the translational start site of *ndrg1a* (*Figure 1—figure supplement 1A*) and the human β-globin gene (standard, control MO) (*Schmajuk et al., 1999*) were commercially obtained (Gene Tools, LLC) and administered at 2 nl/embryo using 0.1 mM working solution. Each MO was injected at the one- to four-cell stages into the yolk margin just underneath the embryo proper. The MO sequences are as follows:

*ndrg1a* (MO1): 5'- CGTCCATTCTCAACGAGGACACCCG –3'.
Standard control MO: 5'-CCTCTTACCTCAGTTACAATTTATA-3'.

Specificity of the *ndrg1a* MO1 was confirmed using immunolabeling and western blotting, which revealed that Ndrg1a protein was absent or greatly reduced in MO-injected embryos (*Figure 1—figure supplement 1E, G*). As expected, *ndrg1a* transcript levels, detected using whole-mount *in situ* hybridization, appeared normal in *ndrg1a* MO-injected embryos *Figure 1—figure supplement 1F*.

## Whole-mount *in situ* hybridization

Embryos were fixed in 4% paraformaldehyde (PFA) at the desired stages and processed for wholemount *in situ* hybridization as previously described (*Thisse and Thisse, 2014*), but with a modified anti-DIG antibody pre-absorption solution that contains fish powder. The following primers were used to synthesize sense and antisense probes. The underlined font indicates the T7 RNA polymerase binding site.

### Antisense atp1a1a1.2:

FWD: 5'-CTTATGGAATCCAGGTGGCATC-3'

REV: 5'-<u>TAATACGACTCACTATAG</u>GCAGTACCTTCAACACAGTTGG-3'

## Sense atp1a1a1.2:

FWD: 5'-<u>TAATACGACTCACTATAG</u>CTTATGGAATCCAGGTGGCATC-3'
REV: 5'-GCAGTACCTTCAACACATTGG-3'

## Antisense kcnj1a.1:

FWD: 5'-CCTGGAGGTGTGGCTTTGAT-3'
REV: 5'-<u>TAATACGACTCACTATAG</u>AGCCATGCCATCGAGGAAAA-3'

## Sense kcnj1a.1

FWD: 5'-<u>TAATACGACTCACTATAG</u>CCTGGAGGTGTGGCTTTGAT-3'
REV: 5'-AGCCATGCCATCGAGGAAAA-3"

## Antisense ndrg1a:

FWD: 5'-AGCTCACACCTCCGAAAACC-3'
REV: 5'-<u>TAATACGACTCACTATAG</u>ACGATGTCTCTGTGCTGCAT-3'

## Sense ndrg1a:

FWD: 5'-<u>TAATACGACTCACTATAG</u>AGCTCACACCTCCGAAAACC-3'
REV: 5'-ACGATGTCTCTGTGCTGCAT-3'

## Antisense slc12a3:

FWD: 5'-ATGGGAATCCAAGGCCCAAG-3'
REV: 5'-<u>TAATACGACTCACTATAG</u>ACACAGGAGCCAATGGTAGC-3'

## Sense slc12a3:

FWD: 5'-<u>TAATACGACTCACTATAG</u>ATGGGAATCCAAGGCCCAAG-3'
REV: 5'-ACACAGGAGCCAATGGTAGC-3'

## Antisense slc20a1a:

FWD: 5'-ACCATCTTTGAGACAGTGGGTG-3'
REV: 5'-<u>TAATACGACTCACTATAG</u>CAGCCCAGTGAGATTAGCAGGG-3'

## Sense slc20a1a:

FWD: 5'-<u>TAATACGACTCACTATAG</u>ACCATCTTTGAGACAGTGGGTG-3'
REV: 5'-CAGCCCAGTGAGATTAGCAGGG-3'

## Immunolabeling and other labeling procedures

### Immunolabeling

Embryos were fixed in 4% paraformaldehyde (PFA) at 4 °C overnight and washed in 1 x PBS for 30 min. Fixed embryos were permeabilized with cooled acetone for 5 min at –20 °C, followed by a 5 min wash in 1 x PBS. Embryos were subsequently incubated in Inoue blocking solution (*Inoue and Wittbrodt, 2011*) 5% Normal goat Serum (NGS) (Abcam, cat# ab7481, lot# GR325285-5), 2% BSA (Fisher Scientific, cat# BP1600-100, lot# 196941), 1.25% Triton X-100 (Fisher Scientific, cat#

BP151-500, lot# 172611) in 1 x PBS for 1 hr at room temperature (RT) on a rotating platform (80 RPM). Incubation in primary antibodies was performed in I-buffer solution (1% normal goat serum, 2% bovine serum albumin, 1.25% Triton X-100, in 1 x PBS) for 2 days at 4 °C on a rotating platform (80 RPM). Embryos were then washed three times, 30 min each, with 1 x PBS at room temperature. Secondary antibodies, diluted in I-buffer, were applied for 1 day at 4 °C on a rotating platform (80 RPM). After secondary antibody incubation, embryos were washed three times with 1 x PBS for 30 min.

*Primary antibodies*: anti-NDRG1 @ 1:500 (Sigma Aldrich, catalog # HPA006881, lot# A69409, rabbit polyclonal), anti-ATP1A1A @ 1:500 (DHSB, catalog # a5s, lot# 10/17/19, mouse monoclonal), anti-GFP @ 1:500 (Invitrogen, cat# A21311, lot# 2207528, rabbit polyclonal, Alexa Fluor 488 conjugated), anti-ZO-1 @ 1:500 (Invitrogen, cat#40-2200, lot# UB280595, rabbit polyclonal). *Secondary antibodies*: Goat anti-rabbit Alexa Fluor 594 (Abcam, cat# ab150080, lot# GR3373513-1), Goat anti-mouse Alexa Fluor 488 @ 1:500 (Abcam, catalog # ab150113, Lot#GR3373409-1). *Other labels added after secondary antibodies*. Phalloidin-Alexa 594 (Invitrogen, cat#A12381, lot# 2256805), Phalloidin-Alexa 405 (Invitrogen, cat#A30104, lot# 2277734) and DAPI (Thermo Scientific, cat# 62248, lot# WD3246961) were used according to the manufacturer's instructions.

## Whole-mount proximity ligation assay (PLA)

To identify *in situ* protein-protein interactions, PLA was performed using the PLA Duolink kit (Sigma Aldrich, catalog# DUO92101). The protocol was performed according to the manufacturer's instructions with the following modifications: All steps were performed in 96-well plates, 40 μl final volume, at RT, rotating (80 RPM). Embryos were incubated in primary antibody solution at RT overnight. Embryos were then treated with Duolink PLA probes at RT overnight, followed by incubation in the ligation solution at RT overnight and an additional 1 hr at 37 °C. Between primary antibody, Duolink PLA probe, and ligation solution incubations, samples were washed with 1 x Wash Buffer A at RT for 3x20 min. Embryos were incubated in the amplification solution at RT overnight, followed by 1 hr at 37 °C. Lastly, samples were washed in 1 x Wash Buffer B at RT for 3x20 min, then in 0.01 x Wash Buffer B at RT for 20 min.

## Sectioning and microscopy

### Sectioning embryos

Transverse sections (40 μm) of fixed and labeled embryos were obtained using a vibratome (Vibratome, 1500). Embryos were mounted in agarose (4 %) blocks prior to sectioning. Sections through the anterior and posterior pronephric duct were distinguished based on tissue morphology, as the distance between the two tubules is larger in the anterior region. The average distance between anterior pronephric ducts is 94.3 μm while the distance between posterior pronephric ducts is on average 48.1 μm.

### Compound microscopy

Live, dechorionated embryos were immobilized using the sedative tricaine methanesulfonate at 0.001% w/v in embryo medium E3 (5 mM NaCl, 0.33 mM CaCl2, 0.17 mM KCl, 0.33 mM MgSO4). Sedated or fixed embryos were mounted on depression slides in E3 medium. Lateral views of 24–48 hpf embryos were viewed at ×50 magnification using an Axioskop II compound microscope (Carl Zeiss, Axioscope II) and imaged with an AxioCam (Carl Zeiss, 504 CCD camera).

### Confocal microscopy

For lateral views, 24–48 hpf embryos were mounted in E3 medium on glass-bottom culture dishes with size #1.5 coverslips (Mattek, cat# P50GC-1.5–14 F). Images were captured with a 5 X air objective lens using an inverted SP5 laser scanning confocal microscope (Leica Microsystems). For imaging transverse sections of embryos, a 63 X oil objective lens was used. Images were analyzed and processed using FIJI ImageJ (NIH). When comparing protein levels between embryos in the same experimental repeat, all specimens were processed using identical settings, including gain.

## Western blot analysis

Embryos were dechorionated and mechanically lysed in precooled cOmplete Protease Inhibitor Cocktail (Millipore Sigma, catalog# 11836145001). The bicinchoninic acid (BCA) protein assay was used to determine the total protein concentration in embryo lysates. Western blot analysis was performed by loading 7.5 g of total protein per lane into 7.5% Tris-glycine SDS-PAGE gels (Bio-Rad, catalog #4561026) and transferring to methanol-activated PVDF membranes (Bio-rad, cat# 1620177). Specific protein detection was performed using diluted *primary antibodies* as follows: anti-NDRG1 @ 1:5000 (Sigma-Aldrich, catalog# HPA006881, rabbit polyclonal); anti-GAPDH @1:5000 (GeneTex, catalog #GTX124503; batch: RRID:AB_11165273, rabbit polyclonal). The *secondary antibody* was used as follows: HRP-linked Goat anti-rabbit @ 1:10,000 (Cell Signaling Technology, catalog #7074; RRID:AB_2099233). Molecular weights were determined using pre-stained protein ladders (Thermo Fisher Scientific, catalog #26616; GE Healthcare, catalog #RPN800E). Protein abundance was detected on membranes using the SuperSignal West Femto enhanced chemiluminescence kit (Thermo Fisher Scientific, catalog #34095) and Blu-C high-contrast autoradiography film (Stellar Scientific, catalog #BLC-57–100).

## Drug treatments

Sodium azide (NaN3). (catalog #S2002-5G) was used at 300 µM concentration in embryo medium (E3) to increase endogenous lactate levels, as previously described (*Oginuma et al., 2017*). Since sodium azide arrests or delays development, age and stage-matched untreated controls were used.

Protease inhibitor MG-132 (EMD Millipore Corp, cat# 474791–5 MG, lot# 3537249) and autophagy inhibitor Chloroquine (Sigma Aldrich, cat# C6628-50G, batch # 0000090940) were used at 100 µM and 1 mM, respectively in E3.

Ouabain octahydrate (Millipore Sigma, cat# PHR1945-300MG, lot# LRAB2762) was used at 3 mM in E3.

## Hypoxia and anoxia treatments

For all hypoxia experiments, oxygen levels were controlled using a PLAS LABS' Controlled Environmental Chamber (model# 856-HYPO). A combination of pure nitrogen (N2) (Airgas) and room air was perfused into the chamber and resulting oxygen concentrations were measured using the PLAS LABS' built-in oxygen sensor. Additionally, a handheld portable oxygen meter (Yellow Springs Instruments (YSI) Item# 626972) was placed inside of the hypoxia chamber to directly monitor oxygen levels in E3 embryo medium. E3 medium was placed in the PLAS LABS chamber in a beaker and aerated using an air pump to accelerate equilibration of dissolved oxygen levels with the chamber's ambient oxygen level for 12 hr before experimentation.

## Survival tests

### Anoxia survival

Survival was assayed immediately following anoxia exposure. We initially identified parameters indicative of a 'dead phenotype' by setting aside anoxia-exposed embryos with different levels of tissue damage and scoring for survival post-reoxygenation. Based on these trials, we established that the characteristics of dead embryos include: opaque appearance, tissue damage, swollen body with extensive edema, and detached yolk. Thus embryos exhibiting one or more of these characteristics were classified as dead or dying and were not used in subsequent analyses.

### Anoxia + re-oxygenation survival

Embryos that survived anoxia were transferred to normoxic water and examined after two days of re-oxygenation for signs of viability, including heartbeat and response to touch. The surviving embryos were further categorized into either mild or severe edema cases.

## Kidney clearance assay

Kidney function was examined in WT and *ndrg1a*$^{-/-}$ mutants using the kidney clearance assay (KCA) (*Christou-Savina et al., 2015*).

## KCA using embryos raised under normoxic conditions

WT embryos and $ndrg1a^{-/-}$ mutants were injected at 48 hpf in the pericardial cavity with rhodamine dextran (10,000 MW; Invitrogen, cat# D1824, lot# 2270252; working concentration of 5 mg/ml in distilled water). Fluorescence in the heart chamber of injected embryos was imaged and quantified 3 and 24 hpi using an Axioskop II compound microscope (Carl Zeiss) and AxioCam 504 CCD camera (Carl Zeiss). Embryos were imaged at 50 X total magnification using identical exposure settings. Images were taken in an area of interest centered on the heart and analyzed and processed using ImageJ (NIH).

## KCA using embryos exposed to anoxia followed by re-oxygenation

Same procedure as described above but with the following modifications: 24 hpf WT embryos and $ndrg1a^{-/-}$ mutants were exposed to 12 h of anoxia, followed by two days of re-oxygenation, then injection of rhodamine dextran (under normal oxygen).

### ATP assay

24 hpf WT and $ndrg1a^{-/-}$ mutant embryos were exposed to 0, 6, 12 hr of anoxia and snap-frozen along with stage-matched control embryos in liquid nitrogen immediately post-anoxia and removal of E3 medium. Metabolites were extracted and ATP concentration (mM/embryo) measured using a luminescent ATP Detection Assay kit (Abcam, catalog # ab113849), according to the manufacturer's instructions. For the experiments involving Ouabain, 24 hpf WT embryos were exposed to 12 hr of anoxia or 12 hr of anoxia + Ouabain in E3 medium and snap-frozen in liquid nitrogen and processed as described above.

### Lactate assay

24 hpf WT and $ndrg1a^{-/-}$ mutant embryos were exposed to increasing durations of either anoxia or sodium azide (300 μM) and snap-frozen along with stage-matched untreated embryos in liquid nitrogen immediately post-anoxia or azide treatment and removal of E3 medium. The L-lactate fluorometric kit (BioVision, catalog #K607) was used according to the manufacturer's instructions for small biological samples.

### Isothermal calorimetry

ITC experiments were carried out using a MicroCal ITC200 (Malvern Panalytical Ltd; Malvern, UK). The raw ITC data were extracted using Origin 7 software. The temperature in the instrument was set to 28.5 °C. The reference power was set to 5 μcal s−1, and the initial delay after the equilibration was set at 240 s. The feedback mode/gain setting was set to high and the equilibration option was set to 'fast equilibration & auto'. In addition, the ITC experiments were performed without the injection syringe and stirring. Due to the MicroCal iTC200 apparatus requirements, the injection syringe parameter was set as follows: Total # of injections: 1; Syringe Concentration: 1 mM; Cell concentration: 1 mM; Stirring Speed: 100 RPM; Injection volume: 2 μL; Duration: 4 s; Spacing: 5000 s. Prior to embryo injection into the sample cell, the cell was filled with 200 μL of anoxic (0% O2) 1 x E3 medium. Pure water was used in the reference cell. Each experiment began with the equilibration of the ITC and an initial baseline recording for at least 4 min. Either 15 WT or $ndrg1a^{-/-}$ 24 hpf embryos pre-exposed to 24 hr of anoxia (0% O2) or without anoxia treatment (normoxia control) were injected into the sample cell using a pre-measured, and pre-cut P200 micropipette tip within 1 min after equilibration and establishment of an initial baseline recording. The embryos were allowed to sink down into the sample cell, following which the excess E3 medium outside of the sample cell was removed using a Hamilton syringe.

### Metabolomic analysis

A metabolomic analysis was performed on 5 hpf WT embryos exposed to 1 hr of anoxia and age-matched normoxic controls (5 hpf). A total of 400 μl of methanol and 400 μl of water were added as a polar solvent to decant the embryos over ice. After decanting, 400 μl of isopropanol was added as a nonpolar solvent. Samples were rocked at 4 ° C for 20 min and centrifuged at 13,000 g for 5 min. The water-soluble metabolites (top of phase separation) was extracted and stored at - 80 °C. Four biological replicates were obtained and 10 embryos were used per sample. Mass spectra were acquired using a Phenomenox Luna NH2 column connected to an Agilent 1200 HPLC and Agilent 6210

ESI-TOF mass spectrometer in a negative ion mode. Data were analyzed with Agilent software packages (MassHunter, Mass Profiler, Profinder and Mass Profiler Professional) using 1000 ion counts as an arbitrary cutoff. An extracted ion chromatograph (EIC) was then generated for each ion. Ions that did not show clear peaks were discarded from the subsequent analysis. Ions were identified by comparing molar mass against Mzed database (*Draper et al., 2009*). The lactate peak (observed: 89.0240, calculated mass for M-H ions is 89.0239) was extracted, and the area was used for comparison.

## Alignment of NDRG family members

The protein sequences of several vertebrate NDRG family members (human: NDRG3, NP_114402.1; mouse: NDRG3, NP_038893.1, zebrafish: Ndrg1a, NP_998513.2, zebrafish: Ndrg1b: NP_956986.2, zebrafish Ndrg3a: NP_955811.1, zebrafish Ndrg3b: XP_005162043.1) were aligned using the Clustal Omega Multiple Sequence Alignment software (https://www.ebi.ac.uk/Tools/msa/clustalo/) to identify homologous regions of interest.

## Measurements

### Fluorescence intensity measurements of the pronephric duct

Fluorescence intensity corresponding to ATP1A1A protein levels in the anterior and posterior pronephric duct was measured using lateral views of immunolabeled embryos. Two regions of interest (ROI) centered along the anterior or posterior halves of the pronephric duct were delineated for intensity measurements using ImageJ (NIH).

### Fluorescence intensity measurements of ionocytes

Fluorescence intensity corresponding to ATP1A1A protein levels in ionocytes was measured by drawing a tight ROI around individual ionocytes (5-10) in the yolk ball and flank region demarcated by the distal ends of the yolk extension using ImageJ (NIH).

## Experimental design and statistical analysis

Graphing and statistical analysis were performed using Prism 9 (Graph Pad). Statistical significance was declared in circumstances where $p < or = 0.05$. Experiments comparing three or more groups were analyzed using a two-way ANOVA test. An unpaired t-test was used when comparing means of two independent groups. Data were reported as the mean +/- SEM unless otherwise indicated.

## Acknowledgements

We acknowledge M Van Doren, D Eisenmann, S Miller, P Robinson and B Weinstein for their helpful comments on our manuscript. We also acknowledge T de Carvalho and S Larson for help with imaging during the COVID19 pandemic. In addition, access to ITC facilities in the HHMI lab at UMBC (Michael F Summers) is gratefully acknowledged. This project was supported by funding from the Department of Defense (W81XWH-16-1-0466) and the National Institute of Health/NICHD (R21HD089476) to R Brewster and Y.-S Lee. We thank T Addo for contributing to the analysis of *ndrg1a* function. A Gabel, B Canales and A Osei-Ntansah were supported by a National Institute of Health /NIGMS MARC U*STAR (T34 HHS 00001) National Research Service Award to UMBC. S Viswanathan was supported in part by a grant to UMBC from the Howard Hughes Medical Institute through the HHMI Adaptation Project. Polina Kassir was supported by Experimental Learning Award through the Sondheim Public Affairs Scholars Program.

## Additional information

### Funding

| Funder | Grant reference number | Author |
|---|---|---|
| U.S. Department of Defense | W81XWH-16-1-0466 | Young-Sam Lee Rachel Brewster |

| Funder | Grant reference number | Author |
|---|---|---|
| National Institutes of Health | R21HD089476 | Young-Sam Lee<br>Rachel Brewster |
| National Institutes of Health | T34 HHS 00001 | Austin M Gabel<br>Afia Osei-Ntansah<br>Bryanna Canales |
| Howard Hughes Medical Institute | | Austin M Gabel<br>Afia Osei-Ntansah<br>Bryanna Canales<br>Soujanya Viswanathan |
| National Institute of Health | T34 GM136497 | Soujanya Viswanathan |

The funders had no role in study design, data collection and interpretation, or the decision to submit the work for publication.

### Author contributions

Jong S Park, Conceptualization, Data curation, Formal analysis, Supervision, Validation, Investigation, Visualization, Methodology, Writing – original draft, Project administration, Writing – review and editing; Austin M Gabel, Formal analysis, Validation, Investigation, Visualization, Methodology, Writing – review and editing; Polina Kassir, Neil D Tran, Soujanya Viswanathan, Formal analysis, Validation, Investigation, Visualization; Lois Kang, Formal analysis, Validation, Investigation, Methodology; Prableen K Chowdhary, Afia Osei-Ntansah, Validation, Investigation, Visualization; Bryanna Canales, Formal analysis, Validation, Investigation; Pengfei Ding, Resources, Formal analysis, Supervision, Validation, Investigation, Visualization, Methodology; Young-Sam Lee, Resources, Data curation, Formal analysis, Supervision, Funding acquisition, Investigation, Visualization, Methodology, Project administration, Writing – review and editing; Rachel Brewster, Conceptualization, Resources, Data curation, Formal analysis, Supervision, Funding acquisition, Visualization, Project administration, Writing – review and editing

### Author ORCIDs

Jong S Park http://orcid.org/0000-0001-7989-4630
Rachel Brewster http://orcid.org/0000-0002-9058-2045

### Ethics

This study was conducted using protocols approved by the Institutional Animal Care and Use Committee at the University of Maryland, Baltimore County (IACUC protocol: # RB011081821; UMBC OLAW Assurance number: D16-00462). The IACUC approves the use of animals in this research as it meets the standards described in the NIH Guide for the Care and Use of Laboratory Animals and the "Animal Welfare Act."

### Decision letter and Author response

Decision letter https://doi.org/10.7554/eLife.74031.sa1
Author response https://doi.org/10.7554/eLife.74031.sa2

---

# Additional files

### Supplementary files

• Transparent reporting form

### Data availability

Source Data files have been provided for the following figures (see figure legends or source data legends for details): Figure 1-figure supplement 1: G Figure 1-figure supplement 3: D Figure 2: A, B, C, E Figure 3: C Figure 3-figure supplement 1: B Figure 4: B Figure 5: E, F, G Figure 5-figure supplement 1: A-C Figure 6: A-C, D Figure 7: A, C, E.

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
