## [Editor Report]

The manuscript details the function of the N-Myc downstream-regulated gene 1 (NDRG1) during induced hypoxia using the anoxic developing zebrafish as a model system. With some additional support for the central claim of a switch for metabolic suppression, this paper will be of interest to scientists with a focus on kidney development, factors that regulate hypoxic survival, and metabolism in response to stress conditions.

---

## [Decision Letter]

**Decision letter after peer review:**

Thank you for submitting your article "N-myc Downstream Regulated Gene 1 (NDRG1) functions as a molecular switch for cellular adaptation to hypoxia" for consideration by *eLife*. Your article has been reviewed by 2 peer reviewers, and the evaluation has been overseen by a Reviewing Editor and Didier Stainier as the Senior Editor. The reviewers have opted to remain anonymous.

Essential revisions:

1) A weakness is lack of direct data to characterize "metabolic suppression" and evidence linking NKA degradation and metabolic suppression to organismic adaptation and survival against hypoxia. Can the authors clarify role of NKA abundance on metabolic suppression. Furthermore, the parameters used to define metabolic suppression needs to be clarified. As of now the data seems to support the role of NDRG1a in regulating NKA abundance.

2) There was questions that had to do with methodology and statistics. In some cases the overall sample size was small for what is expected within the field (N and n), there were questions about hybridization technique, proper mounting of the embryos, stage of development (may need to note somite number), and use of controls as noted by reviewers. For example, regarding Figure 6D although anoxia induces arrest, may need to determine if some level of development occurred and quantify lactate level with normoxia control (same developmental stage). For several of the experiments/graphs the statistical methods used and P values were not provided. Review each graph and note method and significant differences.

3) Reviewers point out several conclusions that need to be toned down given that the results do not completely support the conclusion provided.

*Reviewer #1 (Recommendations for the authors):*

1. It would help strengthen the paper if the authors could clarify the contribution of changes in NKA abundance in metabolic suppression and organismic survival against hypoxia. If the authors claim this regulatory axis does play a causal role, it would be helpful to consider whether genetic or pharmacological manipulation of NKA or ATP1A1A can affect organismic survival against anoxia/hypoxia in wt and ndrg1a mutants. The best experiment would be testing a genetically modified and degradation resistant atp1a1a allele, or an allele of ndrg1a that specifically interfaces with NKA, but it remains challenging since the underlying molecular mechanism of NKA degradation by ndrg1a is largely unknown.

2, The authors used "metabolic suppression" throughout the paper, but did not have a clear definition described or specific metabolic parameters measured. Quite unexpectedly, ATP levels as authors showed actually increased at 6 hrs after anoxia, and ndrg1a dependent changes of ATP levels did not occur until 12 hrs after anoxia though NKA degradation was already evident at 6 hrs. At what time points and by what measures do the author mean "metabolic suppression"? What explains the discrepancy between ATP and NKA levels at 6 hrs?

3, The authors used the Whole-mount proximity ligation assay (PLA) assay to infer protein-protein interaction between NDRG1a and NKA. While this assay convincingly showed these two are in very close proximity in vivo, it does not exclude the possibility that they interact indirectly via other proteins. The authors may need to tone down and clarify this point, or verify whether they indeed bind to each other directly, e.g. by in vitro biochemistry and ectopic expression in heterologous systems (Y2H etc).

*Reviewer #2 (Recommendations for the authors):*

Across the manuscript the stage of embryos in the assays is not performed correctly. Supplemental figure 1 is a prime example as the larvae analyzed for ndgr1 expression clearly do not match one another. For example the std mo is much younger that the WT and the morphant is older. the conclusions therefore cannot be adequately supported by the different stages.

The *in situ* hybridizations are plagued by significant background and for many genes, appropriate probes/images have been published showing much less head staining. For example see images for slc20a1a. Critical conclusions in Figure 1 are drawn from the *in situ* hybridizations and they are not overall of good quality, equivalent stages, have high background, and do not include sense controls. Also in Figure 1 the ATP1a1a protein is thought to be equal but clearly looks increased in mutants. It requires some quantification and likely more embryos. In addition, the sagittal mounting is inconsistent across the paper. Some are appropriate but others are curved on a different angled plane complicating analysis. They should be performed again.

In Figure 2E, the mutants already have edema, this assay would be more accurate if you compare mutants without edema with wildtype without edema.

Figure 3a2 the image is not mounted correctly. In many of the fluorescent images it can be noted that the yolk or yolk extension of different samples is unique. Or that the magnification area/region seems different. Also in the survival assays there is a mention of the scanning for necrosis, however no details in this area are provided in the methods and it is only vaguely addressed.

In other images of Figure 3 the regions and zoom appear to be different. On b4, you can clearly see the otic placode but on the other images you cannot. Are these taken at the same resolution. the quantification in C is also unclear. What is it normalized too? Is there statistical analysis performed at individual time points for (C). The transverse sections are also difficult to know where and what region we are looking and if they are equivalent sections.

The increased expression of ATP1a1 is a conclusion that is inconsistent with Figure 1. Specifically, situ hybs in Figure 1, where kidney development and the PD were concluded to be normal. However kcnj1a.1 ionocytes also look to be increased in mo and mut.

Can there be an explanation regarding how embryos are genotyped for assays such as the ATP.

This manuscript is intriguing and of interest, however the figures lack sense controls, have high background, are not mounted appropriately, and in many cases the age of the embryos is in question. These changes are likely required to substantiate the authors claims.

---

## [Author Response]

Essential revisions:1) A weakness is lack of direct data to characterize "metabolic suppression" and evidence linking NKA degradation and metabolic suppression to organismic adaptation and survival against hypoxia. Can the authors clarify role of NKA abundance on metabolic suppression. Furthermore, the parameters used to define metabolic suppression needs to be clarified. As of now the data seems to support the role of NDRG1a in regulating NKA abundance.

Metabolic suppression is a survival response to preserve energy that is activated by environmental stressors such as low oxygen or lack of food. Mechanisms underlying this response include reorganization of metabolic priorities for ATP expenditure, suppression of energy-demanding cell functions (e.g. ion pumping, transcription, translation, cell cycle arrest) and changes in gene expression that support hypometabolism (Storey and Storey, 2007). In the zebrafish embryo, prolonged exposure to zero oxygen (anoxia) triggers a hypometabolic state that is manifested by developmental and physiological (heartbeat, response to touch) arrest (Padilla and Roth, 2001). The signaling mechanisms that trigger this organism-level response are currently unknown, but lactate is a candidate proximal signal (Lee, 2021). We have now more clearly defined metabolic suppression on page 3-4 of the Introduction.

To clarify, we did not mean to imply that Ndrg1a-mediated NKA degradation is the sole axis of the anoxia response. Our model (p 34 of the Results section in the original manuscript -initial submission), which we have now moved to the Discussion section on page 26, is that Ndrg1a-mediated NKA downregulation and apparent concomitant reduction in ATP hydrolysis may contribute to hypometabolism. However, metabolic suppression in the zebrafish embryo is an early and organism-wide response, while Ndrg1a expression is spatially restricted and its activation is delayed relative to the onset of anoxia. These findings suggest that Ndrg1a activation is likely to be triggered downstream of hypometabolism.

To address the concern of Reviewer 1 that we did not directly measure metabolic suppression, we developed isothermal calorimetry (ITC) as an assay to quantify heat flux in the intact zebrafish embryo, in response to anoxia. Heat dissipation is closely tied to the activity of NKA and other energy-demanding processes and hence is an indirect measure of hypometabolism (Buck et al., 1993). Using ITC, we now show that heat dissipation is decreased in WT and *ndrg1a^-/-^* mutant embryos exposed to prolonged anoxia, relative to their respective normoxic controls (Figure 5—figure supplement 1). These data support the model that metabolic suppression occurs in response to anoxia and is not dependent on Ndrg1a activity in the kidney. However, we noted that ITC values measured for *ndrg1a^-/-^* mutants were lower than for WT embryos, suggestive of a disruption in energy homeostasis. The ITC experiment is described in Material and Methods and the ITC data can be found in the Results section, page 19 and Figure 5—figure supplement 1.

Our ability to directly test the contribution of NKA degradation in the kidney to hypoxia tolerance was hampered by a lack of experimental tools. To our knowledge, there are no *atp1a1a* mutations or drugs that prevent NKA degradation. Furthermore, repetition of measurements of total ATP in *ndrg1a* mutants to increase sample size numbers (as requested by Reviewer 2) did not support the earlier conclusion that NKA degradation conserves energy (Figure 5F), most likely because ATP generated in kidney cells is a small fraction of total ATP levels. That said, NKA downregulation is a conserved response to hypoxia that has been linked to energy conservation, and hence is thought to contribute to enhanced survival; at least in hypoxia-tolerant organisms (Bogdanova et al., 2005; Bogdanova et al., 2016; Buck and Hochachka, 1993). We now confirm that Ouabain treatment (12 hours) of WT zebrafish embryos exposed to prolonged anoxia can rescue global ATP levels to values observed for 6 hours of anoxia exposure (Figure 5G), confirming that NKA inhibition preserves energy under hypoxia.

We thus propose that NKA downregulation may play a beneficial role in hypoxia tolerance but have toned down the strength of these conclusions in the Discussion section on page 26.

2) There was questions that had to do with methodology and statistics. In some cases the overall sample size was small for what is expected within the field (N and n), there were questions about hybridization technique, proper mounting of the embryos, stage of development (may need to note somite number), and use of controls as noted by reviewers. For example, regarding Figure 6D although anoxia induces arrest, may need to determine if some level of development occurred and quantify lactate level with normoxia control (same developmental stage). For several of the experiments/graphs the statistical methods used and P values were not provided. Review each graph and note method and significant differences.

Methodology. Reviewer 2 had questions about the method used to assess survival using necrosis as a readout – in particular as this method pertains to the selection of embryos for the ATP assay. Our methodology, which is now described in Material and Methods on page 41, entailed initially identifying parameters indicative of a “dead phenotype”, by setting aside anoxia-exposed embryos with different levels of tissue damage and scoring for survival post-reoxygenation. Characteristic of dead embryos include: opaque appearance, tissue damage, swollen body with extensive edema and detached yolk. Thus embryos exhibiting one or more of these characteristic were classified as dead or dying and were not used in subsequent analyses.

Quantification and Statistics. Reviewer 2 had questions about the quantification in Figure 3C. To clarify, measurements were normalized to the highest fluorescence intensity value in the entire set. We also provided (previously) the 2-way ANOVA analysis for both C1 (anterior) and C2 (posterior) data sets. In fact, all graphs and figures with p-values were supplemented with a description of statistical methods in the figure legend, along with corresponding p-values and the full statistical analysis was compiled in an excel spreadsheet.

Sample size. We agree that some sample sizes were on the smaller end, in particular those for the *in situ* hybridization experiments. However, in some cases, the sample size was adequate but we listed the number of embryos *imaged* rather than the total number of embryos processed and visually scored without imaging. Since we repeated the *in situ* hybridization procedure to improve the quality of the images (Figure 1 and Figure 1—figure supplement 2), we also increased the sample size for those experiments. Sample size was also increased for the total ATP measurements, which resulted in a change in our conclusion (Figure 5F and Results page 18). For other experiments, we went back to our spreadsheets and included, where appropriate, all specimens that were qualitatively analyzed, whether micrographs were generated or not. These edits appear in Figure legends.

Staging and imaging issues.

Quality. To improve the quality of the images, we used an improved wholemount ISH procedure that reduces background label. We further ensured that control embryos were stage-matched to experimental embryos and that all specimens were mounted properly and imaged at the same magnification (Figure 1A and Figure 1—figure supplement 2). With regards to Figure 3B specifically, we have provided a new set of images for the IF experiment to ensure consistency between the area of the yolk extension that is in view and confirmed that the specimens in panel Figure 3B were imaged at the same resolution and light intensity.

Controls for *in situ* hybridization. We have now provided images of embryos labeled with control sense riboprobes in Figure 1—figure supplement 2. These embryos have little to no label, suggesting that the signal generated with anti-sense probes in Figure 1A is specific.

Issue with lactate experiment – Figure 6D*.* A concern was raised about whether differential lactate levels between normoxic controls and anoxia-treated experimental groups reflect variation in developmental stage rather than response to anoxia exposure. We agree that this is a potential issue given that we used age-matched (24 hpf) rather than stage-matched (27 hpf) normoxic controls. We have now included stage-matched controls and do not observe significant differences in lactate levels relative to the age-matched experimental groups. These data are shown in Figure 6D. In addition, we have also included stage-matched controls for the ATP assay with additional experiments (Figure 5F).

3) Reviewers point out several conclusions that need to be toned down given that the results do not completely support the conclusion provided.

We have edited our Abstract, Results (pages 18, 19) and Discussion (page 26) to provide a more conservative interpretation of the data.

Reviewer #1 (Recommendations for the authors):1. It would help strengthen the paper if the authors could clarify the contribution of changes in NKA abundance in metabolic suppression and organismic survival against hypoxia. If the authors claim this regulatory axis does play a causal role, it would be helpful to consider whether genetic or pharmacological manipulation of NKA or ATP1A1A can affect organismic survival against anoxia/hypoxia in wt and ndrg1a mutants. The best experiment would be testing a genetically modified and degradation resistant atp1a1a allele, or an allele of ndrg1a that specifically interfaces with NKA, but it remains challenging since the underlying molecular mechanism of NKA degradation by ndrg1a is largely unknown.

We agree with Reviewer 1 that our experimental design does not directly test the contribution of NKA downregulation to hypoxia adaptation. However, to our knowledge, there are no *atp1a1a* mutations or drugs that prevent NKA degradation. We did attempt to test whether Ouabain, an inhibitor of all NKA α-subunit isoforms, could rescue lethality of *ndrg1a* mutants exposed to prolonged anoxia, but the broad range activity of this drug obscured data interpretation.

Repetition of measurements of total ATP in *ndrg1a* mutants to increase sample size numbers did not support the earlier conclusion that NKA degradation conserves energy (Figure 5F), most likely because ATP generated in kidney cells is a small fraction of total ATP levels. That said, NKA downregulation is a conserved response to hypoxia that has been linked to energy conservation, and hence is thought to contribute to enhanced survival; at least in hypoxia-tolerant organisms (Bogdanova et al., 2005; Bogdanova et al., 2016; Buck and Hochachka, 1993). Furthermore, we now show that Ouabain treatment (12 hours) of WT zebrafish embryos exposed to prolonged anoxia can rescue global ATP levels to values observed for 6 hours of anoxia exposure (Figure 5G), confirming that NKA inhibition preserves energy under hypoxia.

We thus propose that NKA downregulation may play a beneficial role in hypoxia tolerance but have toned down the strength of these conclusions in the Discussion section on page 26.

2, The authors used "metabolic suppression" throughout the paper, but did not have a clear definition described or specific metabolic parameters measured.

Metabolic suppression is now defined in the Introduction on page 3-4, as a survival response to preserve energy that is activated by environmental stressors such as low oxygen or lack of food. Mechanisms underlying this response include reorganization of metabolic priorities for ATP expenditure, suppression of energy-demanding cell functions (e.g. ion pumping, transcription, translation, cell cycle arrest) and changes in gene expression that support hypometabolism (Storey and Storey, 2007).

To address the concern that we did not directly measure metabolic suppression, we developed isothermal calorimetry (ITC) as an assay to quantify heat flux in the intact zebrafish embryo, in response to anoxia. Heat dissipation is closely tied to the activity of NKA and other energy-demanding processes and hence is a measure of hypometabolism (Buck et al., 1993). Using ITC, we now show that heat dissipation is decreased in WT and *ndrg1a^-/-^* mutant embryos exposed to prolonged anoxia, relative to their respective normoxic controls (Figure 5—figure supplement 1). These data support the model that metabolic suppression occurs in response to anoxia and is not dependent on Ndrg1a activity in the kidney. However, we noted that ITC values measured for *ndrg1a^-/-^* mutants were lower than for WT embryos, suggestive of a disruption in energy homeostasis. The ITC experiment is described in Material and Methods and the ITC data can be found in the Results section, page 19 and Figure 5—figure supplement 1.

Quite unexpectedly, ATP levels as authors showed actually increased at 6 hrs after anoxia, and ndrg1a dependent changes of ATP levels did not occur until 12 hrs after anoxia though NKA degradation was already evident at 6 hrs. At what time points and by what measures do the author mean "metabolic suppression"? What explains the discrepancy between ATP and NKA levels at 6 hrs?

We currently do not fully understand why ATP levels increase after 6 hours of anoxia (Figure 5F), however this response is observed in both WT and *ndrg1a^-/-^* mutants and is therefore Ndrg1a-independent. One possible explanation is that, at this time point, there is a residual pool of ATP generated from oxidative-phosphorylation and an increase in ATP production from glycolysis that combine to produce this spike. In addition, Ndrg1a-independent ATP-conserving responses may be activated prior to 6 hours of anoxia that further contribute to the increase in ATP levels.

With regards to the timing of events, PLA reveals an interaction between Ndrg1a and NKA beginning at 6 hours in the posterior kidney, which coincides with the onset of NKA degradation. We can unfortunately not comment on when ATP preservation occurs given that the assay to measure total ATP levels was not sensitive enough to detect changes that occur specifically in the kidney and ionocytes across all samples tested.

3, The authors used the Whole-mount proximity ligation assay (PLA) assay to infer protein-protein interaction between NDRG1a and NKA. While this assay convincingly showed these two are in very close proximity in vivo, it does not exclude the possibility that they interact indirectly via other proteins. The authors may need to tone down and clarify this point, or verify whether they indeed bind to each other directly, e.g. by in vitro biochemistry and ectopic expression in heterologous systems (Y2H etc).

We used PLA in our study to interrogate the *in situ* interaction between Ndrg1a and NKA. Consistent with our findings, others have shown that these two proteins physically interact in prostate cancer cells, using co-IP and mass spec analysis (Tu et al., 2007). That said, we acknowledge the limitations of PLA to demonstrate direct physical interaction and have toned down our interpretation of these data (p 15).

Reviewer #2 (Recommendations for the authors):Across the manuscript the stage of embryos in the assays is not performed correctly. Supplemental figure 1 is a prime example as the larvae analyzed for ndgr1 expression clearly do not match one another. For example the std mo is much younger that the WT and the morphant is older. the conclusions therefore cannot be adequately supported by the different stages.

To address this concern, we performed a new round of wholemount *in situ* hybridization on age-matched embryos and confirmed that the spatial distribution of *ndrg1a* is the same across all experimental groups (WT, STD morpholino (MO)-injected, *ndrg1a* MO-injected and *ndrg1a*^-/-^). Please refer to Figure 1—figure supplement 1A, Figure 1.

The *in situ* hybridizations are plagued by significant background and for many genes, appropriate probes/images have been published showing much less head staining. For example see images for slc20a1a. Critical conclusions in Figure 1 are drawn from the *in situ* hybridizations and they are not overall of good quality, equivalent stages, have high background, and do not include sense controls. also in Figure 1 the ATP1a1a protein is thought to be equal but clearly looks increased in mutants. It requires some quantification and likely more embryos. In addition, the sagittal mounting is inconsistent across the paper. Some are appropriate but others are curved on a different angled plane complicating analysis. They should be performed again.

We have improved the quality of the images by repeating the wholemount *in situ* hybridization procedure for all probes along with sense controls, and using a more effective anti-DIG pre-absorption protocol to reduce background. Given the large number of panels, we chose to only include the images of embryos labeled with the antisense probe in Figure 1 but we provide images for the sense probe-labeled embryos in Figure 1—figure supplement 2.

In Figure 2E, the mutants already have edema, this assay would be more accurate if you compare mutants without edema with wildtype without edema.

Following prolonged anoxia some *ndrg1a* mutants exhibited edema while others had normal morphology. To address the concern raised by Reviewer 2, we compared the kidney clearance of WT embryos to mutants with or without edema following prolonged anoxia exposure. Interestingly, the following RFI values (measurement of kidney filtration capacity, with higher values indicating lower kidney filtration) were obtained for the different experimental groups – from the highest to lowest RFI values: *ndrg1a* mutants without edema (64.1% RFI), *ndrg1a* mutants with edema had (51.9% RFI), and WT (44.3% RFI). These data indicate that both mutant groups (with or without edema) had worse filtration capacity than WT embryos but the mutants with edema in fact appeared to maintain better kidney filtration capacity post-anoxia than that of mutants without edema.

Figure 3a2 the image is not mounted correctly. In many of the fluorescent images it can be noted that the yolk or yolk extension of different samples is unique. Or that the magnification area/region seems different. Also in the survival assays there is a mention of the scanning for necrosis, however no details in this area are provided in the methods and it is only vaguely addressed.In other images of Figure 3 the regions and zoom appear to be different. On b4, you can clearly see the otic placode but on the other images you cannot. Are these taken at the same resolution.

We have now adjusted Figure 3 to ensure consistency between the area of the yolk extension that is in view and confirm that the specimens in panel B were imaged with the same resolution and light intensity.

The quantification in C is also unclear. What is it normalized too? Is there statistical analysis performed at individual time points for (C).

The quantification in Figure 3C is normalized to the highest fluorescence intensity value in the entire set. We performed a 2-way ANOVA analysis for both C1 (anterior) and C2 (posterior) data sets. Please refer to the attached excel sheets with 2-way ANOVA analysis.

The transverse sections are also difficult to know where and what region we are looking and if they are equivalent sections.

We have included an illustration of where the anterior and posterior pronephric duct (PD) sections are located in Figure 1—figure supplement 3E. In addition, we have made quantitative measurements of both the anterior and posterior PD sections from the center of the left PD to the center of the right PD to set length criteria for determining anterior vs posterior PD sections. The average distance between anterior pronephric ducts is 94 microns while the distance between posterior pronephric ducts is on average 48 microns. (Excel file for PD length quantification provided, see Author response tables 1 and 2).

**Author response table 1. sa2table1:** Distance between anterior pronephric tubules.

Image	Trial	Distance	Conversion to micron
Snap15	1	2.005	119.06091
	2	2.029	120.486078
	3	2.003	118.942146
Snap63	1	1.804	107.125128
	2	1.835	108.96597
	3	1.814	107.718948
Snap137	1	1.888	112.113216
	2	1.889	112.172598
	3	1.892	112.350744
Snap252	1	1.142	67.814244
	2	1.142	67.814244
	3	1.153	68.467446
Snap261	1	1.296	76.959072
	2	1.278	75.890196
	3	1.264	75.058848
Snap304	1	1.391	82.600362
	2	1.381	82.006542
	3	1.377	81.769014
			
Average		1.58794	94.29505308

**Author response table 2. sa2table2:** Distance between posterior pronephric tubules.

Image	Trial	Distance	Conversion to micron
Snap20	1	0.799	47.446218
	2	0.813	48.277566
	3	0.815	48.39633
Snap96	1	0.859	51.009138
	2	0.85	50.4747
	3	0.889	52.790598
Snap126	1	0.681	40.439142
	2	0.669	39.726558
	3	0.687	40.795434
Snap237	1	0.979	58.134978
	2	0.969	57.541158
	3	0.959	56.947338
Snap282	1	0.94	55.81908
	2	0.926	54.987732
	3	0.928	55.106496
Snap329	1	0.62	36.81684
	2	0.593	35.213526
	3	0.616	36.579312
			
Average		0.81067	48.13920594

The increased expression of ATP1a1 is a conclusion that is inconsistent with Figure 1. Specifically, situ hybs in Figure 1, where kidney development and the PD were concluded to be normal. However kcnj1a.1 ionocytes also look to be increased in mo and mut.

Our conclusion may not have been stated clearly, leading to some confusion. We proposed that ATP1A1A is regulated by Ndrg1a at the post-translational rather than transcriptional level (via lysosomal or proteasomal degradation); hence, in *ndrg1a* morpholino-injected embryos and mutants, *atp1a* mRNA level is normal (Figure 1) but ATP1A1A protein level is high relative to WT controls (Figure 3). With respect to *kcnj1a.1* expression in ionocytes, the apparent increase in signal may be due to the high level of background in the initial images. Having repeated the wholemount *in situ* hybridization with a new blocking reagent that reduces background, *kcnj1a.1* expression level is similar across all experimental groups (refer to revised Figure 1A).

Can there be an explanation regarding how embryos are genotyped for assays such as the ATP.

Our methodology, which is now described in Material and Methods on page 41, entailed initially identifying parameters indicative of a “dead phenotype”, by setting aside anoxia-exposed embryos with different levels of tissue damage and scoring for survival post-reoxygenation. Characteristic of dead embryos include: opaque appearance, tissue damage, swollen body with extensive edema and detached yolk. Thus embryos exhibiting one or more of these characteristic were classified as dead or dying and were not used in subsequent analyses.

For the ATP assay, both WT and *ndrg1a* mutant embryos are quickly examined under the bright field objective (in anoxic water) for the criteria described above and immediately snap-frozen in dry ice following the removal of excess solution.

**References**

Buck, L. T., Hochachka, P. W., Schön, A., & Gnaiger, E. (1993). Microcalorimetric measurement of reversible metabolic suppression induced by anoxia in isolated hepatocytes. *The American journal of physiology*, *265*(5 Pt 2), R1014–R1019. https://doi.org/10.1152/ajpregu.1993.265.5.R1014Bogdanova, A., Grenacher, B., Nikinmaa, M., & Gassmann, M. (2005). Hypoxic responses of Na+/K+ ATPase in trout hepatocytes. *The Journal of experimental biology*, *208*(Pt 10), 1793–1801. https://doi.org/10.1242/jeb.01572Bogdanova, A., Petrushanko, I. Y., Hernansanz-Agustín, P., & Martínez-Ruiz, A. (2016). "Oxygen Sensing" by Na,K-ATPase: These Miraculous Thiols. *Frontiers in physiology*, *7*, 314. https://doi.org/10.3389/fphys.2016.00314Lee T. Y. (2021). Lactate: a multifunctional signaling molecule. *Yeungnam University journal of medicine*, *38*(3), 183–193. https://doi.org/10.12701/yujm.2020.00892Padilla, P. A., & Roth, M. B. (2001). Oxygen deprivation causes suspended animation in the zebrafish embryo. Proceedings of the National Academy of Sciences of the United States of America, 98(13), 7331–7335. https://doi.org/10.1073/pnas.131213198Storey, K. B., & Storey, J. M. (2007). Tribute to P. L. Lutz: putting life on 'pause'--molecular regulation of hypometabolism. *The Journal of experimental biology*, *210*(Pt 10), 1700–1714. https://doi.org/10.1242/jeb.02716Tu, L. C., Yan, X., Hood, L., & Lin, B. (2007). Proteomics analysis of the interactome of N-myc downstream regulated gene 1 and its interactions with the androgen response program in prostate cancer cells. *Molecular & cellular proteomics:MCP*, *6*(4), 575–588. https://doi.org/10.1074/mcp.M600249-MCP200